# Modulation of occluding junctions alters the hematopoietic niche to trigger immune activation

**Rohan J Khadilkar, Wayne Vogl, Katharine Goodwin, Guy Tanentzapf\***

Department of Cellular and Physiological Sciences, University of British Columbia, Vancouver, Canada

**Abstract** Stem cells are regulated by signals from their microenvironment, or niche. During *Drosophila* hematopoiesis, a niche regulates prohemocytes to control hemocyte production. Immune challenges activate cell-signalling to initiate the cellular and innate immune response. Specifically, certain immune challenges stimulate the niche to produce signals that induce prohemocyte differentiation. However, the mechanisms that promote prohemocyte differentiation subsequent to immune challenges are poorly understood. Here we show that bacterial infection induces the cellular immune response by modulating occluding-junctions at the hematopoietic niche. Occluding-junctions form a permeability barrier that regulates the accessibility of prohemocytes to niche derived signals. The immune response triggered by infection causes barrier breakdown, altering the prohemocyte microenvironment to induce immune cell production. Moreover, genetically induced barrier ablation provides protection against infection by activating the immune response. Our results reveal a novel role for occluding-junctions in regulating niche-hematopoietic progenitor signalling and link this mechanism to immune cell production following infection.

DOI: https://doi.org/10.7554/eLife.28081.001

**\*For correspondence:** tanentz@mail.ubc.ca

**Competing interests:** The authors declare that no competing interests exist.

## Introduction

Stem cells are essential for animal development and allow the maintenance and regeneration of tissues. Stem cell fate and behavior is often governed by signals derived from their local microenvironment, also known as the stem cell niche (*Jones and Wagers, 2008*; *Morrison and Spradling, 2008*). When stem cells divide the daughter cells can either self-renew to replenish the stem cell population or differentiate to give rise to a particular cell type (*Fuchs and Chen, 2013*). Stem cell differentiation and self-renewal are tightly regulated as producing too many stem cells can result in tumor formation while producing too many differentiated cells depletes the stem cell population (*He et al., 2009*). Exploring how a niche regulates its resident stem cells is a major topic in stem cell biology. An important biological process where stem cell-niche interactions play a key regulatory role is the process of hematopoiesis, the production of new blood cells from hematopoietic progenitor cells, known as prohemocytes in *Drosophila*. *Drosophila* hematopoiesis produces blood cells, called hemocytes, that have specialized and essential functions in mediating fly immunity.

There are two waves of hematopoiesis in *Drosophila*, the first of which occurs in the embryo. While embryonic hemocytes persist into later stages, a second wave of hematopoiesis takes place in a specialized organ called the Lymph Gland (LG) in larva (*Evans et al., 2003*; *Hartenstein, 2006*; *Holz et al., 2003*; *Márkus et al., 2009*; *Stofanko et al., 2008*; *Makhijani et al., 2011*; *Zaidman-Rémy et al., 2012*; *Leitão and Sucena, 2015*; *Vlisidou and Wood, 2015*). The LG consists of the primary lymph gland lobe and a series of posterior lobes (secondary lymph gland lobes). The primary lobe serves as the main site of hematopoiesis in larvae while the functional importance of the

posterior lobes is currently not well defined (*Jung et al., 2005*; *Kalamarz et al., 2012*; *Letourneau et al., 2016*). The prohemocytes in the posterior lobes tend to be more proliferative but, similar to the primary lobe prohemocytes, express markers like DE-cadherin and JAK/STAT receptor, Domeless (*Jung et al., 2005*). As the main site of larval hematopoiesis the primary lobe has been the subject of intense study (*Lebestky et al., 2003*; *Jung et al., 2005*; *Krzemień et al., 2007*; *Mandal et al., 2007*; *Sinenko et al., 2009*; *Kulkarni et al., 2011*; *Benmimoun et al., 2012*; *Pennetier et al., 2012*; *Khadilkar et al., 2014*; *Milton et al., 2014*; *Tokusumi et al., 2015*). The primary lobe is organized into three regions: at the posterior end is a well-defined hematopoietic niche (also known as PSC or Posterior Signalling Centre). This niche is connected to the Medullary Zone (MZ), which houses the prohemocytes, while the differentiated hemocytes are found in the Cortical Zone (CZ) (*Figure 1a*). Prohemocytes can differentiate into three blood cell types: plasmatocytes, crystal cells and lamellocytes (*Evans et al., 2003*; *Jung et al., 2005*), which perform the immune functions of phagocytosis, melanization, and encapsulation, respectively. The cells of the PSC express unique markers and form a morphologically distinct structure (*Lebestky et al., 2003*; *Krzemień et al., 2007*; *Mandal et al., 2007*; *Kulkarni et al., 2011*; *Pennetier et al., 2012*)

The PSC provides a micro-environment that contributes to LG homeostasis, possibly by acting as a source for signals that are essential for regulating the production of differentiated hemocytes (*Mandal et al., 2007*; *Krzemień et al., 2007*; *Pennetier et al., 2012*; *Benmimoun et al., 2015*; *Oyallon et al., 2016*). The signalling environment in the lymph gland is highly complex, with multiple pathways acting simultaneously to regulate hematopoiesis including Wingless, Hedgehog, JAK/STAT, Dpp and Notch (*Lebestky et al., 2003*; *Krzemień et al., 2007*; *Mandal et al., 2007*; *Sinenko et al., 2009*; *Pennetier et al., 2012*; *Dey et al., 2016*). Mutations that perturb these signalling pathways result in severe disruptions to hematopoiesis including loss of the prohemocytes or hematopoietic malignancies (*Crozatier and Vincent, 2011*). Most of what is known about cell signalling in the LG is derived from work on the primary lobes, where cell-signaling pathways have been extensively characterized (*Lebestky et al., 2003*; *Krzemień et al., 2007*; *Mandal et al., 2007*; *Sinenko et al., 2009*; *Kulkarni et al., 2011*; *Benmimoun et al., 2012*; *Pennetier et al., 2012*; *Khadilkar et al., 2014*; *Milton et al., 2014*; *Tokusumi et al., 2015*). However, there is also some evidence that perturbation of the signals that regulate prohemocyte behavior in the primary lobe also affect hemocyte differentiation in the secondary lobes (*Crozatier et al., 2004*; *Krzemień et al., 2007*; *Tan et al., 2012*; *Benmimoun et al., 2015*). The PSC in the primary lobe of the LG is known to be the source of a number of cell signalling ligands including Dpp, Hedgehog and Wingless which are all required for prohemocyte maintenance or for regulating blood cell differentiation (*Mandal et al., 2007*; *Sinenko et al., 2009*; *Pennetier et al., 2012*). Addional signalling molecules, such as Pvr, STAT and ADGF-A are known to act outside of the PSC to regulate prohemocyte maintenance (*Mondal et al., 2011*; *Mondal et al., 2014*; *Khadilkar et al., 2014*). Therefore, ongoing communication between the PSC, the prohemocytes, and differentiating hemocytes is key for maintaining homeostasis in the LG and also for facilitating the ability of prohemocytes to rapidly respond to inductive cues to undergo differentiation (*Krzemień et al., 2007*; *Mandal et al., 2007*; *Mondal et al., 2011*; *Crozatier and Vincent, 2011*; *Khadilkar et al., 2014*; *Mondal et al., 2014*).

Increased production of differentiated hemocytes is one of the main ways by which flies could possibly respond to infection as in the case of wasp infestation (*Krzemień et al., 2007*; *Márkus et al., 2009*; *Sinenko et al., 2011*). There are two arms of the immune response in flies: the humoral arm and the cellular arm. The humoral immune response is largely based on the production of antimicrobial peptides (AMP) while the cellular arm is largely mounted by the hemocytes. Different signalling pathways are activated following immune challenges in *Drosophila*: The Toll and Imd pathways are induced following infection by gram-positive and gram-negative bacteria, respectively (*De Gregorio et al., 2002*; *Ferrandon et al., 2007*), and the EGFR pathway is induced following wasp infestation (*Sinenko et al., 2011*). Wasp infestation is presently the only known immune challenge that affects LG function, since following infestation the PSC induces prohemocyte differentiation (*Crozatier et al., 2004*; *Sinenko et al., 2011*; *Ferguson and Martinez-Agosto, 2014*). It has been shown that the Toll pathway is active in the LG (*Qiu et al., 1998*; *Chiu et al., 2005*), specifically, the PSC responds to wasp infestation by activating the Toll-NFκB signalling (*Gueguen et al., 2013*). However, presently, whether the LG is involved in mounting the immune response following bacterial infection is a major unresolved question. Nonetheless, the ability of the LG to induce

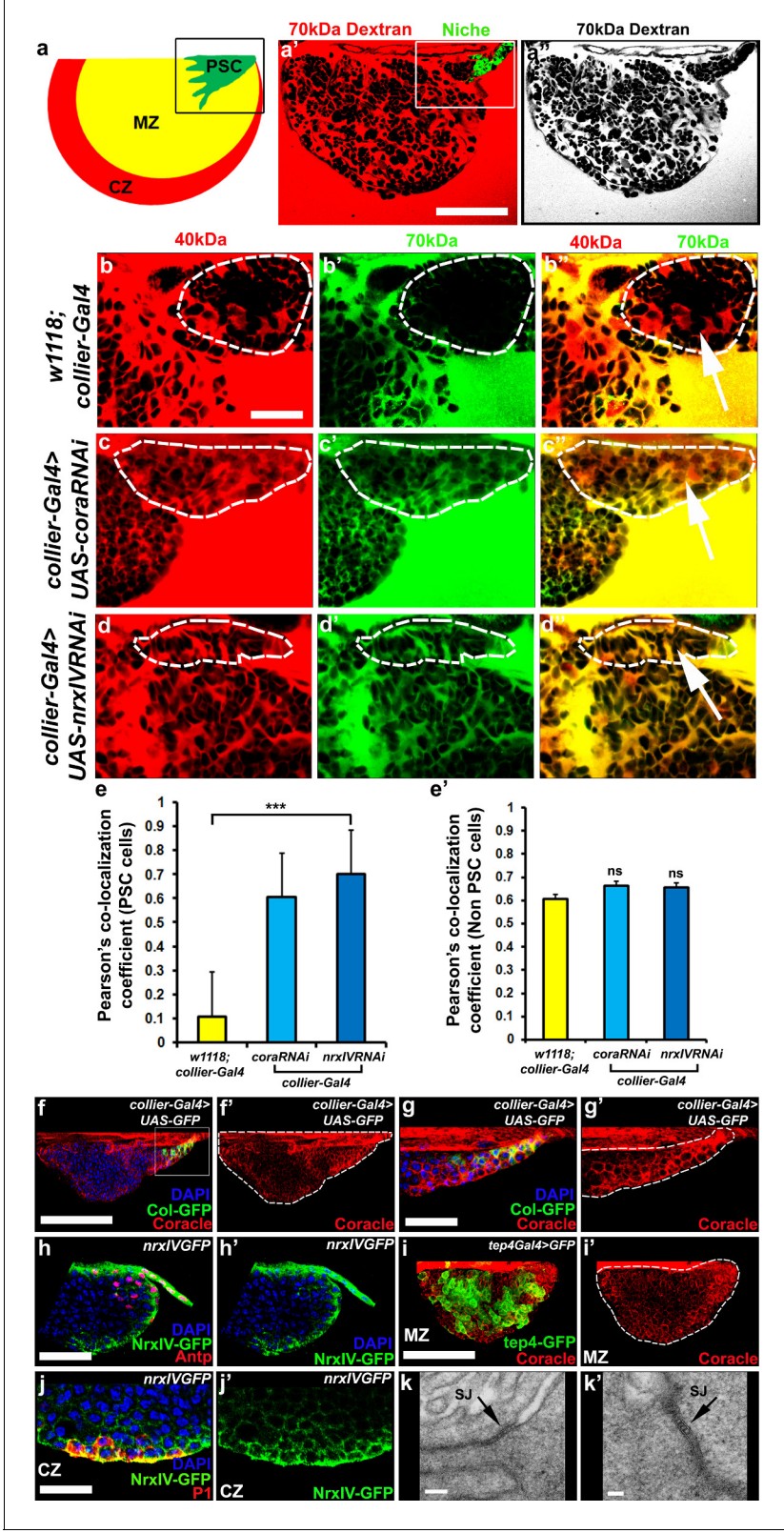

**Figure 1.** Septate junctions form a permeability barrier at the PSC in the lymph gland. (a) Schematic representation of the lymph gland (PSC-Posterior Signalling Centre; MZ-Medullary Zone; CZ-Cortical Zone). (a',a'') 70kDa-dextran (Red) is excluded from the PSC (Green). (b–d'') Dual Dye Assay with 70kDa-dextran (Green) and 40kDa-dextran (Red) dextran in wild-type (b–b'') and following PSC specific knockdown of *cora* or *NrxIV* (c–d''). (e,e') Pearson's co-localization co-efficient quantification of data in b-d in PSC and non-PSC cells. (f–f'') Coracle expression (red) in PSC cells (*collier-*

*Figure 1 continued on next page*

*Figure 1 continued*

*Gal4 >UAS* GFP; green). **(g–g')** Enlarged view of boxed region in **(f)**. **(h–h')**. NrxIV expression (green) in PSC cells (Antp antibody; Red). **(i–i')** Coracle expression (red) in MZ cells (*tep4-Gal4 > UAS* GFP; green). **(j–j')** NrxIV expression (green) in CZ cells (P1 antibody; red). **(k–k')** Electron micrographs showing septate junctions in between PSC cells. Nuclei labeled with DAPI (Blue). **(a'–a'',f,g)** ***=P < 0.001; ns = non significant. Error bars represent s.d. Scale Bars:**(a',a'',f–f', i–i')** 50 μm, **(b–d'',g–h', j–j')** 40 μm, **(k)** 100 nm **(k')** 50 nm.

DOI: https://doi.org/10.7554/eLife.28081.002

The following source data and figure supplements are available for figure 1:

**Source data 1.** Contains numerical quantitation represented in *Figure 1e*.
DOI: https://doi.org/10.7554/eLife.28081.005
**Source data 2.** Contains numerical quantitation represented in *Figure 1e'*.
DOI: https://doi.org/10.7554/eLife.28081.006
**Figure supplement 1.** Low molecular weight dyes are not excluded from the PSC.
DOI: https://doi.org/10.7554/eLife.28081.003
**Figure supplement 2.** Septate junctions in the PSC are absent upon PSC- specific depletion of septate junctions.
DOI: https://doi.org/10.7554/eLife.28081.004

hemocyte differentiation in response to infection is likely to be tied to carefully orchestrated changes in the signalling microenvironment produced by the PSC.

The mechanisms by which prohemocytes are primed towards differentiation and how the cellular immune response becomes activated following infection are not well understood. Stem cell-niche interactions have been shown in multiple systems to be regulated by cell junctions, such as cadherin-based adherens junctions and integrin-based Cell-ECM adhesions (*Tanentzapf et al., 2007*; *Ellis and Tanentzapf, 2010*; *Chen et al., 2013*). Recent evidence has linked occluding junctions, such as vertebrate tight junctions or their invertebrate counterparts Septate Junctions (SJs) (*Banerjee et al., 2006*; *Fairchild et al., 2015*, *2016*), in regulating stem cell-niche interactions. Occluding junctions, serve an essential function in animals by creating permeability barriers that generate different compartment inside and/or in between tissues. For example, in the fly testes an occluding-junction based permeability barrier has a crucial role in regulating stem cell differentiation by enclosing the differentiating germline in an isolated compartment (*Fairchild et al., 2015*). In this system, SJs block external signals from reaching the germline and thus restrict the range of niche-derived stem cell maintenance cues. In the absence of SJs the range of niche-derived stem cell maintenance cues is greatly extended (*Fairchild et al., 2015*). Moreover, SJ mediated regulation of the signalling microenvironment at the niche can control niche size (*Fairchild et al., 2016*). Similarly, occluding junctions have been implicated in the control of intestinal stem cells in *Drosophila* (*Resnik-Docampo et al., 2017*). This raises the question whether occluding junctions might also have a function in regulating prohemocytes in the LG.

Here we demonstrate that SJs play an essential role in regulating hematopoiesis in the *Drosophila* lymph gland. Using a novel dye-based assay we identify a previously uncharacterized permeability barrier separating the PSC from the prohemocytes. This permeability barrier disintegrates following bacterial infection and there is a corresponding change in the signalling microenvironment generated by the PSC. As a result of the change in the PSC-produced microenvironment the differentiation of prohemocytes into immune cells is increased. Importantly, genetic ablation of the permeability barrier around the PSC is sufficient to activate the cellular immune response and provides long-lasting protection against bacterial infection. These results provide new insight into how the PSC responds to bacterial infection in order to initiate the cellular immune response.

## Results

### Septate junctions form a permeability barrier at the PSC in the lymph gland

We recently described the development of a non-invasive dye-based assay to show that the stem cell niche in the *Drosophila* testes was enclosed within a permeability barrier (*Fairchild et al., 2015*). To determine if the LG contained a similar permeability barrier around the PSC we adapted this assay. Dissected LGs were kept in culture and incubated with fluorescently-conjugated dextrans of

various sizes (*Fairchild et al., 2015*). The ability of the dye to access the surface of cells in the LG was assessed and quantified. We found that, in wild-type flies, 10 kDa and 40 kDa dextran freely labeled the surface of all cells (*Figure 1—figure supplement 1a–d'', f–f'*) but that 70 kDa dextran was effectively excluded from PSC (*Figure 1a'–a''*, *Figure 1—figure supplement 1e–e'',f''*). This indicated that a permeability barrier was present at the niche and that the size at which the movement of molecules became restricted was between 40 kDa and 70 kDa. To better, quantitatively, assess permeability barrier function in the LG we further developed this approach. In this enhanced version of the assay, called Dual Dye Assay (DDA), green 70 kDa and red 40 kDa dyes were used simultaneously (see materials and methods). The use of the smaller, non-excluded dye, provides an internal control for changes in the tissue and allows numerical measurement of dye co-localization to assess barrier function. RNAi-mediated knockdown of the essential SJ components Coracle (*cora*) and NeurexinIV (*NrxIV*) in the PSC abolished this permeability barrier (*Figure 1b–d''*). Depletion of Cora or NrxIV in the PSC followed by DDA analysis showed an increase in the Pearson's co-localization coefficient of the 70 kDa and 40 kDa dyes in PSC consistent with failure to exclude the larger, 70 kDa dextran conjugated dye, compared to controls (*Figure 1e–e'*). This effect was specific for the PSC as the ability of the dye to label other regions of the LG were not affected by knockdown of Cora and NrxIV in the PSC (*Figure 1e–e'*). Consistent with the role of Cora and NrxIV in maintaining a permeability barrier in the niche both proteins were enriched in the PSC (*Figure 1f–j'*, *Figure 1—figure supplement 2f–f''*). In addition to this, we also found that ATPα and the *Drosophila* claudins, Sinuous and Kune-kune that are required for pleated septate junction organization and function show high levels of expression in the PSC (*Figure 1—figure supplement 2g–l'*). Moreover, EM analysis revealed the presence of extensive pleated SJs throughout the LG that were especially prominent between cells in the PSC (*Figure 1k–k'*, *Figure 1—figure supplement 2a–c,f*). Since PSC cells do not exhibit the conspicuous polarity of epithelial cells it seems that septate junctions are not localized to a particular domain. Instead our data suggests that in the PSC cells septate junctions are found in multiple sites, corresponding to contact sites with neighboring PSC cells, and form extensive large plaques (*Figure 1k–k'*, *Figure 1—figure supplement 2a–c,f*).

## Loss of Septate Junctions in the PSC results in increased prohemocyte differentiation and a higher number of cells in the PSC

Next, we asked whether and how loss of the permeability barrier impacted the morphology of the LG or the behavior of the prohemocytes. To allay concerns about specificity of expression of the PSC driver (*collier-Gal4*; *Crozatier et al., 2004*) as well as off-target effects of the transgenic RNAi lines used for depleting Cora and NrxIV, additional transgenic RNAi lines and PSC drivers (*Antp-Gal4*, *Mandal et al., 2007*; *Ser9.6-Gal4*, *Bachmann and Knust, 1998*; *Lebestky et al., 2003*) were used to confirm our observations (*Figure 2—figure supplement 3a–n''* and *Figure 2—figure supplement 4a–p*). The efficiency of the knockdown was validated by analyzing the expression of Cora protein subsequent to knockdown (*Figure 2—figure supplement 1h–i''*). Knockdown of Cora or NrxIV resulted in loss of the permeability barrier (*Figure 1*, *Figure 1—figure supplement 1*). Consistent with this, electron micrographs of the PSC region showed that septate junctions were absent following PSC- specific knockdown of SJ components (*Figure 1—figure supplement 2d–e*). Moreover, two additional defects were observed: First, the average number of cells in the PSC was increased (*Figure 2a–d*, *Figure 2—figure supplement 3a''*-c''',g',h-j',n and *Figure 2—figure supplement 4d–f',n*). Second, we observed increased differentiation of the prohemocytes into both the plasmatocyte and crystal cell lineage (*Figure 2e–f', l–m'*, *Figure 2—figure supplement 1a–a'''*, *Figure 2—figure supplement 3d–f''', g''–g''', h''–m', n', n''* and *Figure 2—figure supplement 4g–l',o, p*). To analyze the differentiation of prohemocytes into plasmatocyte and crystal cell lineage quantitatively, we calculated a differentiation index, as reported previously (*Benmimoun et al., 2012*; *Pennetier et al., 2012*; *Morin-Poulard et al., 2016*), by determining the ratio of differentiated hemocytes to the total number of cells in the primary lobe of the LG. Using this approach we observed that the differentiation indices for crystal cells and plasmatocytes were between 5–10 times higher upon knockdown of Cora or NrxIV in the PSC (*Figure 2s,t*).Moreover, we confirmed that none of the lines used for our experiments or as controls contained a mutation in the NimC1 antigen (*Honti et al., 2013*) which would have confounded our observations (*Figure 2—figure supplement 1j–l'*).We also used an additional marker for plasmatocytes, Eater-dsRed, to confirm that this result was due to prohemocyte differentiation (*Tokusumi et al., 2009*) (*Figure 2—figure*

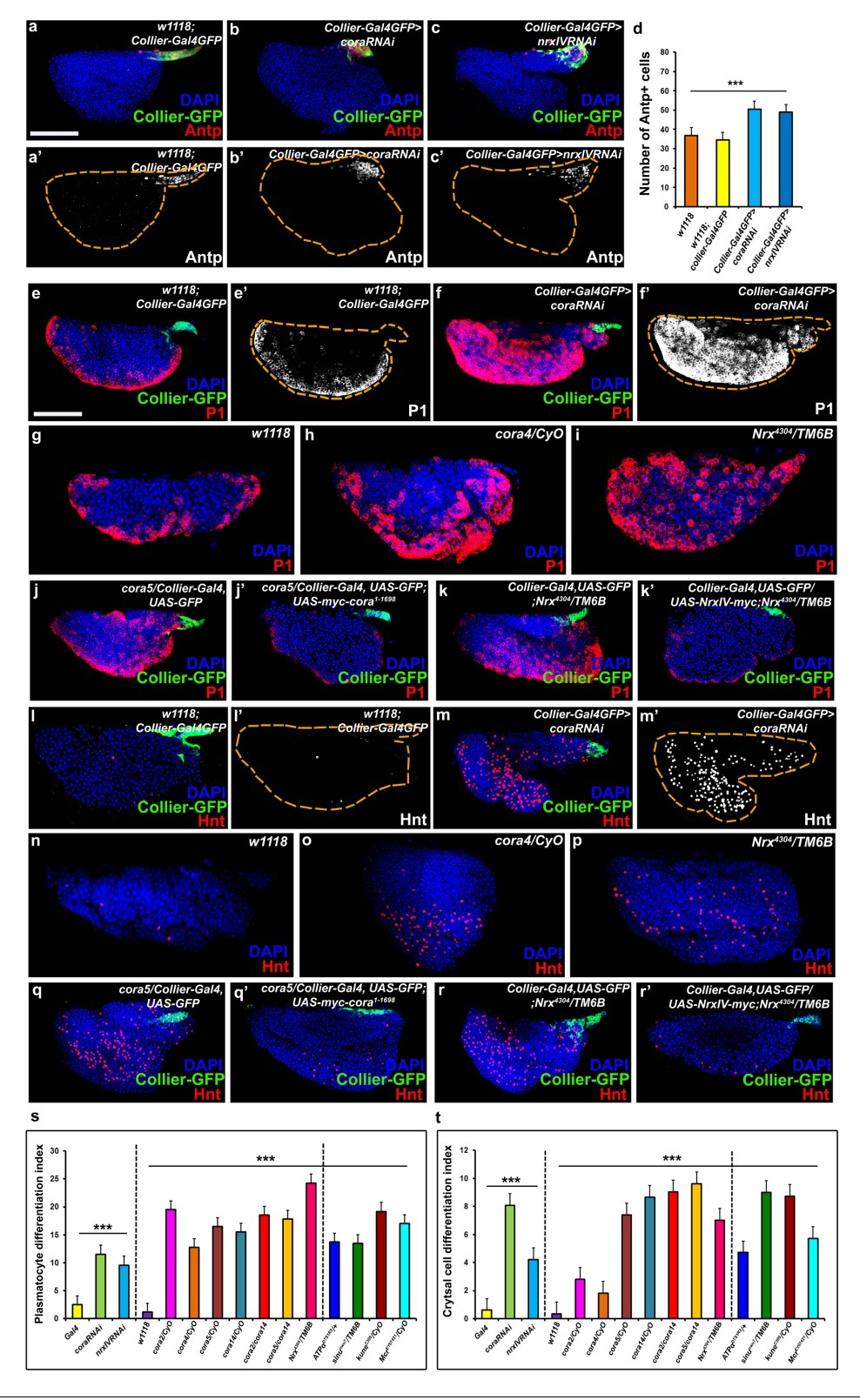

**Figure 2.** The septate junction-mediated permeability barrier is essential for PSC integrity and function. (a–d) PSC (red and green) in wild-type, Cora or NrxIV knockdown LGs, (d) quantification of PSC cell numbers. (e–i) Plasmatocyte differentiation (red, white in e',f') in LG of control (e,e',g), Cora knockdown (f,f'), *cora* (h;cora4/CyO), *Nrx* (i;Nrx[4304]/TM6B) heterozygous mutant alleles. PSC specific rescue of Cora (j') or NrxIV (k') in *cora* or *NrxIV* mutants compared to their respective controls (j,k). (l–p) Crystal Cell differentiation (red; white in l',m') in LG of control (l,l',n), Cora knockdown (m,m'),

*Figure 2 continued on next page*

*Figure 2 continued*

*cora* (o;*cora4/CyO*), *Nrx* (p;*Nrx⁴³⁰⁴/TM6B*) heterozygous mutant alleles. PSC specific rescue of Cora (**q'**) or NrxIV (**r'**) in *cora* or *NrxIV* mutants compared to their respective controls (**q,r**). Quantitation of plasmatocyte (**s**) and crystal cell (**t**) differentiation for LGs with the phenotype shown in panels e-p as well as for a collection of hypomorphic mutations in SJ (septate junction) components as compared to the *w1118* control. (**a–c,e,f,j–k',l,m,q–r'**) PSC labeled with GFP (green; *collier-Gal4 >UAS* GFP) and/or Antennapedia (Red:a-c; white a'-c'). Plasmatocytes labeled with P1 (Red:e,f,g-k';white:e',f'), crystal cells labeled with Hnt (Red:l,m,n-r'; white:l',m'). ***=P < 0.001. Error bars = s.e.m. PSC- specific *Cora* or *NrxIV* knockdown and mutant genotypes were compared to respective parental or wild type controls for the statistical analysis with the t-test. Scale Bar: (**a–c',e–r'**) 50 μm.
DOI: https://doi.org/10.7554/eLife.28081.007

The following source data and figure supplements are available for figure 2:

**Source data 1.** Contains numerical data for plasmatocyte differentiation indices presented in *Figure 2—figure supplement 2s*.
DOI: https://doi.org/10.7554/eLife.28081.013

**Source data 2.** Contains numerical data for crystal cell differentiation indices presented in *Figure 2t*.
DOI: https://doi.org/10.7554/eLife.28081.014

**Figure supplement 1.** Reduced expression of septate junction components affects blood cell homeostasis in the lymph gland.
DOI: https://doi.org/10.7554/eLife.28081.008

**Figure supplement 2.** Depletion of septate junctions in other *Drosophila* organs does not affect lymph gland hematopoiesis.
DOI: https://doi.org/10.7554/eLife.28081.009

**Figure supplement 3.** Depletion of septate junctions using alternate RNAi lines and additional PSC specific Ser9.6-Gal4 driver results in breakdown of permeability barrier and loss of blood cell homeostasis.
DOI: https://doi.org/10.7554/eLife.28081.010

**Figure supplement 4.** Depletion of septate junctions using additional PSC specific Antp-Gal4 results in breakdown of permeability barrier and loss of blood cell homeostasis.
DOI: https://doi.org/10.7554/eLife.28081.011

**Figure supplement 5.** Depletion of septate junctions in the PSC results in a decrease in the proportion of Dome-MESO positive prohemocytes.
DOI: https://doi.org/10.7554/eLife.28081.012

*supplement 1m–n''*). To see if increased hemocyte differentiation resulted in depletion of the prohemocyte population we used the prohemocyte marker Dome-MESO following PSC- specific SJ knockdown (*Figure 2—figure supplement 5a–f*, *Hombría et al., 2005*; *Gao et al., 2009*; *Tokusumi et al., 2009*; *Oyallon et al., 2016*). We found that following SJ knockdown in the PSC, there was an increase in the total number of cells in the primary lobe as well an increase in the number of differentiated hemocytes but a marked decrease in the number of prohemocytes (*Figure 2—figure supplement 5a–f*; *Figure 2—figure supplement 1 Figure 2i'''*). Importantly, even a partial reduction in the levels of SJ components resulted in increased differentiation of prohemocytes. Specifically, a significantly higher differentiation index for plasmatocytes and crystal cells were seen in flies heterozygous for hypomorphic mutations in Cora, NrxIV and other components of SJs (*Figure 2g–i,n–p,s,t* and *Figure 2—figure supplement 1b–f'''*).

## Perturbation of septate junctions outside of the lymph gland does not affect hematopoiesis

Since SJ were depleted in the whole animal in hypomorphic mutations in Cora, NrxIV such phenotypes could result from defects arising outside of the PSC. However, the aberrant prohemocyte differentiation phenotypes were effectively rescued by selectively restoring Cora and NrxIV in the PSC, using PSC-targeted transgene expression in the respective hypomorphic mutant backgrounds (*Figure 2j–k',q–r'*; *Figure 2—figure supplement 1g,g'*). In comparison, in control experiments no such rescue was observed by selectively restoring Cora and NrxIV in the gut using *myo1A-Gal4* (*Morgan et al., 1995*; *Figure 2—figure supplement 1g,g'*, *Figure 2—figure supplement 2a–d*) in their respective hypomorphic mutants. In addition, it is known that LG hematopoiesis is regulated by a variety of systemic factors including Insulin/mTOR mediated nutritional signals as well as neuronal, vascular, and olfactory cues (*Benmimoun et al., 2012*; *Makhijani et al., 2011*; *Morin-Poulard et al., 2016*; *Shim et al., 2013*). To explore the possibility that Cora and NrxIV are involved in mediating these other modes of regulating hematopoiesis in the LG, Cora and NrxIV were also depleted in neuronal tissues (with *elav-Gal4*), the fat body (*lsp2-Gal4*) the gut (*myo1A-Gal4*) (*Morgan et al., 1995*), muscle (*mef2-Gal4*), and salivary gland (*AB1-Gal4*); no significant effects on LG hematopoiesis were observed (*Figure 2—figure supplement 1o–r'*,*Figure 2—figure supplement 2e–r*). Taken

together, these results provide strong support for an essential role of SJs in the PSC in controlling prohemocyte differentiation and LG integrity. Nevertheless, since all the PSC drivers used are also expressed in tissues other than the LG, we cannot fully discount the possibility that effects, mediated outside of the PSC, are also contributing to the phenotypes we observe.

## Over-expression of septate junction components in the PSC affects lymph gland hematopoiesis

Since we see a correlation between depletion of the SJ-mediated permeability barrier at the PSC and defects in hematopoiesis we hypothesized that over-expression of septate junction components might affect hematopoiesis. *collier-gal4* was used to express *UAS-cora* and *UAS-NrxIV* transgenes and overexpression was confirmed using Myc specific antibody to stain the Myc-tagged Cora or NrxIV over-expression lines (*Figure 3a–c'*). No dramatic effects on either crystal cell differentiation or PSC cell numbers were observed (*Figure 3e–e'',f'* and *Figure 3—figure supplement 1a–b*). However, over-expression of NrxIV, but intriguingly not Cora, suppressed plasmatocyte differentiation (*Figure 3d–d'',f*). These results suggest that the levels of SJ proteins in the LG must be carefully controlled to facilitate normal hematopoiesis.

## Bacterial infection disrupts the permeability barrier at the PSC triggering hemocyte differentiation

Bacterial and viral infections are known to cause breakdown of permeability barriers (*Guttman and Finlay, 2009*). We hypothesized that infection would induce an immune response by increasing prohemocyte differentiation and that this may be a direct result of the breakdown of the permeability barrier at the PSC. To test this we infected early third instar *Drosophila* larva with the gram-positive bacteria *B. subtilis* or the gram-negative bacteria *E. coli*. We found that infection, using either

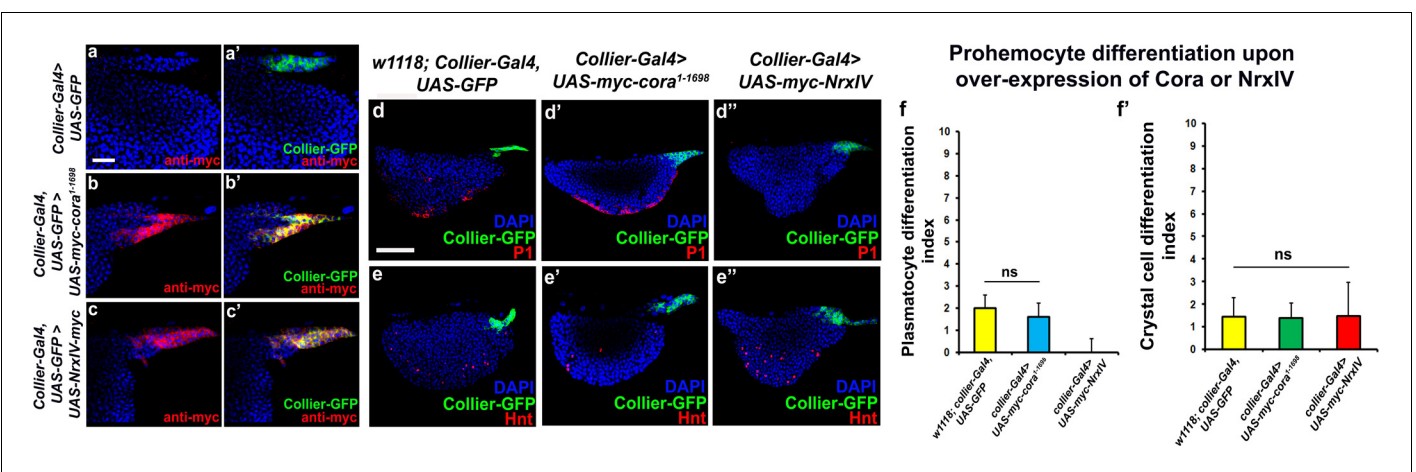

**Figure 3.** Effects of septate junction components overexpression on blood cell homeostasis. (a–c') Validation of myc-tagged Cora or NrxIV over-expression using *collier-Gal4* mediated PSC- specific over-expression using anti-myc tag antibody as compared to its control. P1 positive plasmatocyte counts (d') remain unaltered upon *collier-Gal4* mediated over-expression of Cora whereas they are completely suppressed upon *collier-Gal4* mediated over-expression of NrxIV (d'') as compared to their respective controls (d). Hnt positive crystal cell counts are not affected upon *collier-Gal4* mediated over-expression of Cora (e') or Nrx (e'') as compared to control (e). (f–f') Quantitation of plasmatocyte and crystal cell counts upon *collier-Gal4* mediated over-expression of *cora* or *nrxIV*. PSC is co-labeled in green with *collier-Gal4* driven GFP. Plasmatocytes are labeled with P1 (Red), crystal cells with Hnt (Red). Nuclei are labeled with DAPI (Blue). ns indicates non-significant. Error bars represent s.d. Scale Bar: (a–c') 40 μm, (d–e'') 50 μm.
DOI: https://doi.org/10.7554/eLife.28081.015

The following source data and figure supplement are available for figure 3:

**Source data 1.** Contains numerical data plotted in *Figure 3f*.
DOI: https://doi.org/10.7554/eLife.28081.017
**Source data 2.** Contains numerical data plotted in *Figure 3f'*.
DOI: https://doi.org/10.7554/eLife.28081.018
**Figure supplement 1.** PSC- specific over-expression of Cora or NrxIV does not alter the PSC size.
DOI: https://doi.org/10.7554/eLife.28081.016

feeding or systemic infection models, with either *B. subtilis* or *E. coli* resulted in decreased expression of SJ components in the LG, particularly in the PSC (*Figure 4a–c'*; *Figure 4—figure supplement 1j–l'', m–n'*). We also investigated if the expression of SJ components in other organs like the salivary gland, fat body, foregut or hindgut is affected upon infection and observe that there is no significant difference in the expression levels of SJ components upon systemic infection with *E. coli* (*Figure 4—figure supplement 2a–l*). Importantly, upon infection the permeability barrier at the PSC was abolished (*Figure 4d–f'',g,h*; *Figure 4—figure supplement 1c–f*). Infection with either *B. subtilis* or *E. coli* also led to an increased number of PSC cells (*Figure 4—figure supplement 1a–b''*). Finally, we observed a 4–5-fold increase in prohemocyte differentiation upon infection, using either P1 (*Asha et al., 2003*; *Kurucz et al., 2007*) or Eater-dsRed as plasmatocyte markers (*Tokusumi et al., 2009*) and Hnt as crystal cell marker (*Figure 4i–p*; *Figure 4—figure supplement 1o–r''*; *Benmimoun et al., 2012*). These phenotypes are strikingly similar to those caused by knockdown of SJ components in the PSC. Furthermore, over-expressing SJ components in the PSC prior to infection using *collier-Gal4* abrogated these outcomes; there was no loss of the permeability barrier, as well as no increased differentiation of plasmatocyte or crystal cells and no increase in overall PSC cell numbers (*Figure 5a,b–e* and *Figure 5—figure supplement 1a–h*, *Figure 5—figure supplement 2a–i'*). These results suggest that bacterial infection induces changes in the LG, including prohemocyte differentiation, and that down-regulation in the expression of SJs and the subsequent loss of the permeability barrier is necessary for these changes to occur.

## Flies bearing septate junction depletion in the PSC mount a robust hemocyte- mediated cellular immune response leading to better survival upon infection

It is known that circulating hemocytes contribute to the immune response and promote survival upon oral infection (*Braun et al., 1998*; *Charroux and Royet, 2009*; *Regan et al., 2013*; *Defaye et al., 2009*; *Nehme et al., 2011*). Therefore one possible functional consequence of disruption of the permeability barrier at the PSC, which leads to prohemocyte differentiation, is enhanced protection against infection. Consistent with this, knockdown of SJ components in the PSC conferred a survival advantage to flies infected with either *B. subtilis* or *E. coli* using either a feeding or a systemic infection model (*Figure 4q–r'*; *Figure 4—figure supplement 1s,t*). Moreover, larval infections, that disrupt the permeability barrier, confer a long-term advantage that improves the ability of adult flies to survive bacterial infection (*Figure 4—figure supplement 1i*). We investigated how depletion of SJ components contributes to this survival advantage and found it likely resulted from activation of the cellular arm rather than the humoral arm of the *Drosophila* immune response. To assay the humoral immune response, we tracked the expression of markers for an antimicrobial peptide (AMP) response in the fat body (*Figure 6—figure supplement 1a–l'*). In wild-type controls AMP reporters were only detected following infection; importantly, depletion of SJ components did not induce the endogenous activation of AMP reporters in uninfected controls (*Figure 6—figure supplement 1a–l'*). Intriguingly, over-expressing SJ components in the PSC did not compromise the ability of flies to survive bacterial infection suggesting that other immune mechanisms may take over in such a scenario (*Figure 5—figure supplement 2j*). To assay for activation of the cellular arm we counted the number of circulating hemocytes in early pupal stages 12 hr after puparium formation (APF), shortly after the LG bursts open to release its hemocytes into circulation. Counts were done using two approaches: by incision in the pupal integument and by a whole pupal scraping assay (*Petraki et al., 2015*). Depletion of SJ components in the PSC caused a significant increase in the number of plasmatocytes, crystal cells, and Ad1 positive hemocytes compared to controls whereas lamellocyte counts were not affected (marked with P1 or Eater-dsRed for plasmatocytes, C5 or BcF6GFP or BcF6mcherry for crystal cells, L1 for lamellocytes and Ad1 was used for specifically labeling adult hemocytes (and not embryonic/larval ones), respectively: *Figure 6a–f,g–i'*; *Honti et al., 2014*). This effect was long-lasting as there was a marked increase in the number of adult circulating plasmatocytes and crystal cells as compared to the controls (*Figure 6k–l''*). Moreover, improved survival upon infection in *cora* mutants was likely due to increased number of LG derived circulating hemocytes. This was shown by permanently labeling embryonically derived hemocytes with GFP using a flip-out technique with *gcm-Gal4* to distinguish them from LG (larval) derived hemocytes (*Bataillé et al., 2005*; *Figure 6j–j''*). Embryonic hemocytes are detected by virtue of being dual positive for GFP, driven by *gcm-Gal4*, and dsRed, due to presence of the transgenic plasmatocyte reporter line *Eater-*

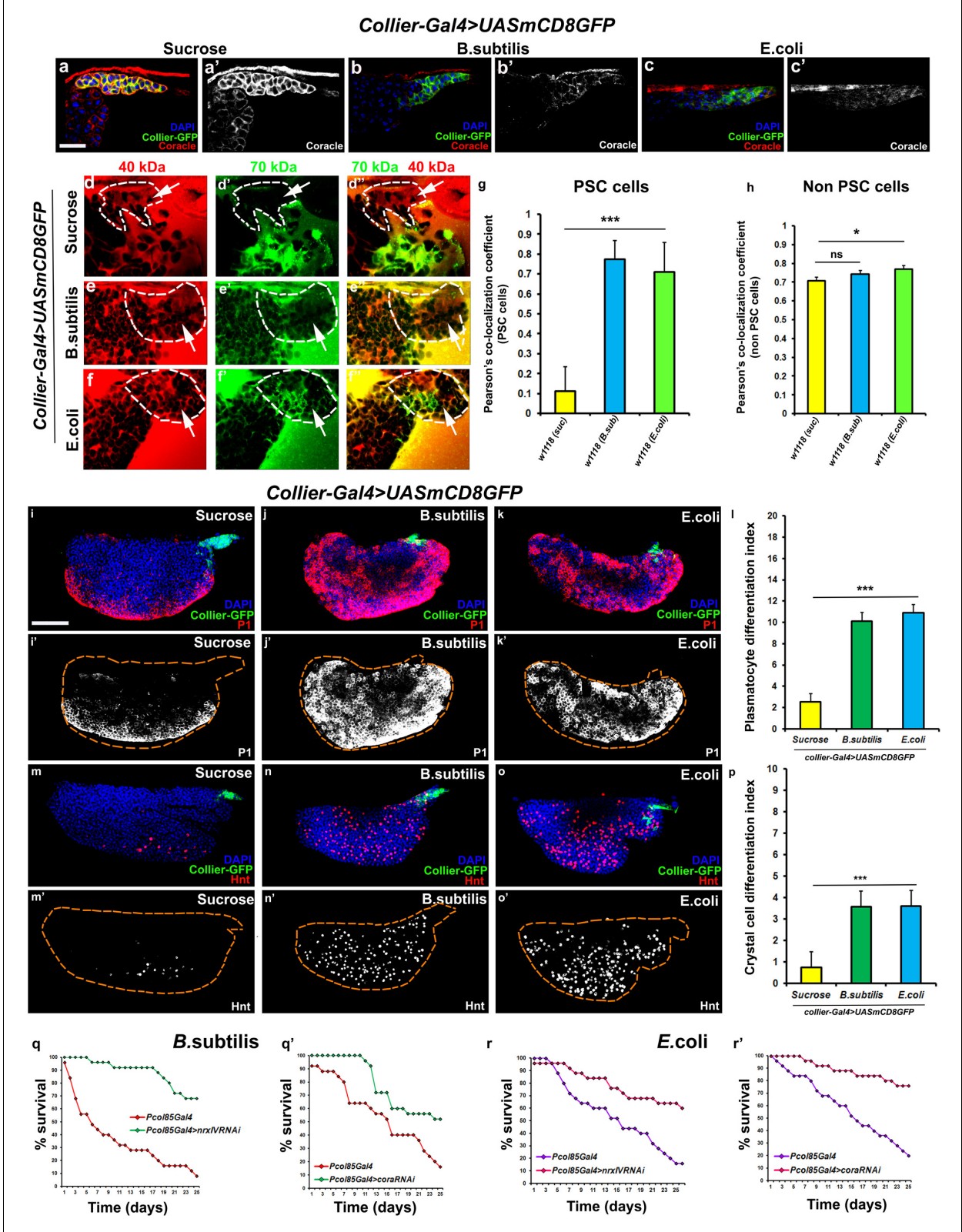

**Figure 4.** Bacterial infection results in breakdown of the permeability barrier leading to prohemocyte differentiation. (a–c') Coracle (Red:a-c; white:a'-c') expression in the PSC (Green) decreased upon infection with (b,b') *B. subtilis* and (c,c') *E. coli* as compared to (a,a') sucrose treated control. (d–h) Dual Dye Assay with 70kDa-dextran (Green) and 40kDa-Dextran (Red) in control, (d–d'') and *B. subtilis* (e–e'') *E. coli* (f–f'') infected larva. (g,h) Pearson's co-localization co-efficient quantification of data in d-h in PSC and non-PSC cells. (i–k) Plasmatocyte differentiation (red; white in i',j',k') in (i) control, and (j)

*Figure 4 continued on next page*

*Figure 4 continued*

*B. subtilis* or (**k**) *E. coli* infected larva. Quantitation of plasmatocyte (**t**) differentiation data. (**m–o**) Crystal cell differentiation (red; white in i',j',k') in (**m**) control, and (**n**) *B. subtilis* or (**o**) *E. coli* infected larva. Quantitation of crystal cell (**p**) differentiation data. (**q–r'**) Survival plots for flies infected with *B. subtilis* (**q,q'**) or *E. coli* (**r,r'**) over time in control versus PSC specific Nrx or Cora knockdown flies. (**a–c,d–l,k–m,o–q**) PSC labelled in green (*collier-Gal4 >UAS* GFP). Plasmatocytes labeled with P1 (Red:i-k; white:i'-k'), crystal cells with Hnt(Red:m-o; white:m'-o'). Nuclei labeled with DAPI (Blue). ***=P < 0.001, *=P < 0.1 and ns = non significant. Error bars represent s.e.m. Scale Bar: (i–k',m–o') 50 μm, (a–f'') 40 μm.

DOI: https://doi.org/10.7554/eLife.28081.019

The following source data and figure supplements are available for figure 4:

**Source data 1.** Contains numerical data plotted in *Figure 4g*.
DOI: https://doi.org/10.7554/eLife.28081.023
**Source data 2.** Contains numerical data plotted in *Figure 4h*.
DOI: https://doi.org/10.7554/eLife.28081.024
**Source data 3.** Contains numerical data plotted in *Figure 4l*.
DOI: https://doi.org/10.7554/eLife.28081.025
**Source data 4.** Contains numerical data plotted in *Figure 4p*.
DOI: https://doi.org/10.7554/eLife.28081.026
**Figure supplement 1.** Bacterial infection induces changes in the prohemocyte microenvironment.
DOI: https://doi.org/10.7554/eLife.28081.020
**Figure supplement 2.** Bacterial infection does not affect Coracle expression in other *Drosophila* organs.
DOI: https://doi.org/10.7554/eLife.28081.021
**Figure supplement 3.** Flies bearing PSC- specific depletion of septate junctions have a better ex-vivo and in-vivo bacterial clearance ability.
DOI: https://doi.org/10.7554/eLife.28081.022

*dsRed.* In comparison, larval (LG) plasmatocytes were only positive for the Eater-dsRed marker. This analysis showed that the population of embryonically derived hemocytes remained fairly stable (*Figure 6j–j''*). Finally, hemocyte activation was also confirmed via another assay, the initiation of filopodial extensions, upon larval bacterial infection (*Regan et al., 2013*) (*Figure 4—figure supplement 1g–h''*).

These results are consistent with the hypothesis that disruption of the permeability barrier at the PSC leads to prohemocyte differentiation and subsequently to a long-lasting activation of the cellular immune response. To test this further, an in-vivo bacterial clearance assay was performed on adult flies, this assay showed that the bacterial load of flies having PSC- specific depletion of SJ components is lesser as compared to the control flies after 24 hr post systemic infection with *E. coli* (*Figure 4—figure supplement 3e*). Furthermore, an ex-vivo bacterial phagocytosis assay demonstrated that the hemocytes derived from adult flies bearing PSC- specific depletion of SJ components had an enhanced ability to phagocytose GFP- tagged *E. coli* as compared to the hemocytes from the controls (*Figure 4—figure supplement 3a–d*). Taken together these data strongly support the notion that the breakdown of the permeability barrier at the PSC changes the ability of flies to fight infection by inducing the cellular arm of the immune response and that these effects are long lasting.

## Activation of toll and imd- mediated innate immune signalling induces breakdown of the permeability barrier at the PSC

Next we asked how infection causes breakdown of the permeability barrier. The Toll and Imd pathways initiate the immune response in *Drosophila* following infection by gram-positive and gram-negative bacteria, respectively (*De Gregorio et al., 2002*). Intriguingly, Toll as well as the Imd pathway component Relish were both detected in the PSC (*Figure 7—figure supplement 1a–f''*). Interestingly, an earlier study has shown that the PSC responds to wasp infestation by activating the Toll-NFκB signalling (*Gueguen et al., 2013*). In order to investigate the possible role of the Toll and Imd pathways in SJ depletion following infection both pathways were ectopically activated in three different ways: in the PSC, in the gut to mimic systemic activation, or ubiquitously throughout the animal. The Toll pathway was activated by expression of a constitutively active Toll transgene, while the Imd pathway was activated by reducing the expression of either of the two Imd inhibitors *Pirk* or *dRYBP*. Activating either the Toll or Imd pathway in the PSC, systemically, or ubiquitously caused loss of the permeability barrier (*Figure 7a–f,j*; *Figure 7—figure supplement 2a–c'*). Consistent with this,

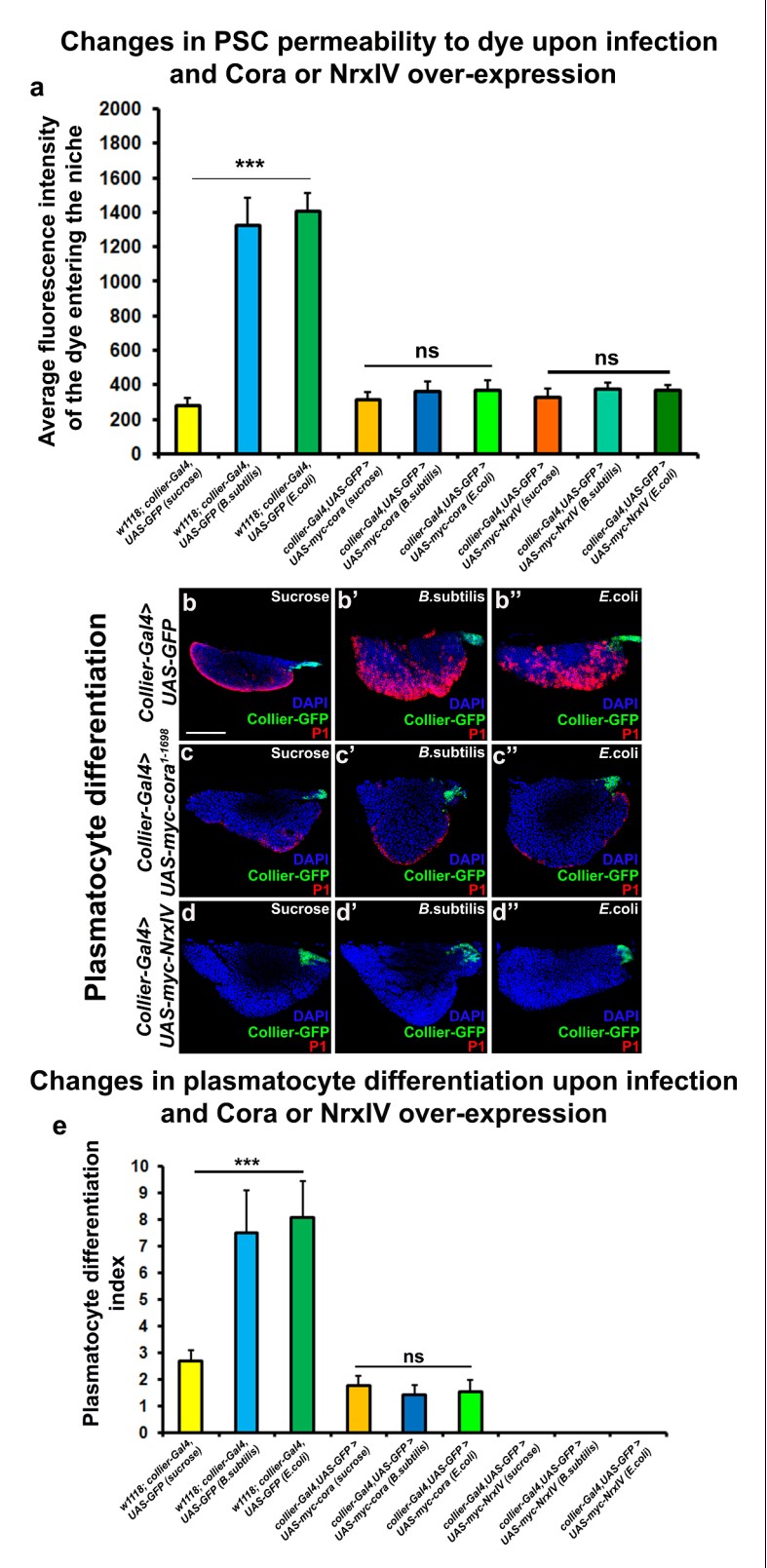

**Figure 5.** Over-expression of septate junction components in the PSC blocks infection-induced permeability barrier breakdown and prohemocyte differentiation. (a) Quantitation of 70 kDa dye influx in the PSC upon infection of larvae over-expressing SJ's in the PSC using *collier-Gal4*. (b–e) In contrast to controls (b–b''), P1-positive plasmatocytes counts are not increased upon *B. subtilis* or *E. coli* mediated infection of larvae over-expressing Cora (c–c'') or NrxIV (d–d'') in the PSC using *collier-Gal4*. (e) Quantitation of plasmatocyte numbers upon infection of larvae over-expressing SJ's in the
*Figure 5 continued on next page*

Figure 5 continued

PSC using *collier-Gal4*. PSC is co-labeled in green with *collier-Gal4* driven GFP. Plasmatocytes are labeled with P1 (Red). Nuclei are labeled with DAPI (Blue). *** indicates p<0.001 and ns indicates non-significant. Error bars represent s.d. Scale Bar: (**b–d''**) 50 µm.

DOI: https://doi.org/10.7554/eLife.28081.027

The following source data and figure supplements are available for figure 5:

**Source data 1.** Contains numerical data for quantitation in *Figure 5a*.
DOI: https://doi.org/10.7554/eLife.28081.030
**Source data 2.** Contains numerical data for quantitation in *Figure 5e*.
DOI: https://doi.org/10.7554/eLife.28081.031
**Figure supplement 1.** Over-expression of septate junctions in the PSC blocks infection induced prohemocyte differentiation.
DOI: https://doi.org/10.7554/eLife.28081.028
**Figure supplement 2.** Over-expression of septate junctions in the PSC blocks infection induced permeability barrier breakdown.
DOI: https://doi.org/10.7554/eLife.28081.029

localized activation of the Toll pathway in the PSC resulted in the down-regulation of Coracle (*Figure 7—figure supplement 1g–h''*). Moreover, activating the Toll and Imd pathways caused the prohemocytes to differentiate into plasmatocytes and crystal cells (*Figure 7g–i'*; *Figure 7—figure supplement 2d–k*). This data is consistent with previous results showing that expressing activated Toll in the larval fat body induces hemocyte differentiation in the LG (*Schmid et al., 2014*). Strikingly, we found that compromising the permeability barrier using mutations in SJ components was able to partially rescue dominant loss-of-function Toll and Imd pathway mutants. Typically flies heterozygous for the Toll and Imd pathway mutants $spz^2ca^1$ and $Rel^{E20}$, respectively, die shortly after infection, but combining them with mutant alleles of *NrxIV* or *cora* provided substantial rescue of this lethality (*Figure 7k*). These data support the idea that the Toll and Imd pathways induce the breakdown of the permeability barrier and that this is an integral part of the mechanism by which the cellular arm of innate immunity is activated by these pathways.

## Septate junction components genetically interact with components of the toll and imd pathway

In further support of the idea that the Toll and Imd pathways act upstream to control permeability barrier function at the PSC, we observed that neither $spz^2ca^1$ nor $Rel^{E20}$ mutants exhibited the characteristic down-regulation of Cora upon infection that was seen in wild-type controls (*Figure 7—figure supplement 1i*). Consistent with our other observations, this inability to inactivate the permeability barrier was accompanied by failure to induce plasmatocyte and crystal cell differentiation or induce hemocyte dispersal in 12 hr APF pupa (*Figure 8b,b',d,e* and *Figure 8—figure supplement 1a,a',c,c',g–h' and m*). To explore in detail the genetic inter-relationship between SJs and the Toll and Imd pathways we investigated double mutants for either $spz^2ca^1$ with *NrxIV* or *cora* and $Rel^{E20}$ with *NrxIV* or *cora*. These double mutants showed induction of both plasmatocyte and crystal cell differentiation and increased hemocyte dispersal in 12 hr APF pupa both before and after bacterial infection (*Figure 8a–a'',c–e* and *Figure 8—figure supplement 1b,b',d–e',f–f'' and i–m*). These data suggest that modifying the permeability barrier around the PSC is a key target of both the Toll and Imd pathways as it provides an effective means of activating prohemocyte differentiation.

## Permeability barrier breakdown at the PSC introduces localized changes in the PSC micro-environment affecting wingless and BMP signalling

Why would loss of the permeability barrier at the PSC cause ectopic prohemocyte differentiation? We envisioned two possible mechanisms whereby a permeability barrier in the LG can shape the PSC signalling microenvironment. The first possible mechanism is by limiting the diffusion of niche-derived signals so that they do not reach the prohemocytes in sufficient concentration. Data consistent with such a mechanism is derived from experiments looking at Dpp signalling, which is primarily active in, and regulates the size of the PSC (*Pennetier et al., 2012*). The transgenic Dpp reporter *Dad-GFP* revealed that the range of BMP (Bone morphogenetic protein) signals was extended further from the PSC following depletion of SJ components (*Figure 9i–k''*). The second possible mechanism is concentrating signals locally, at or near the PSC, by limiting the ability of signals to diffuse

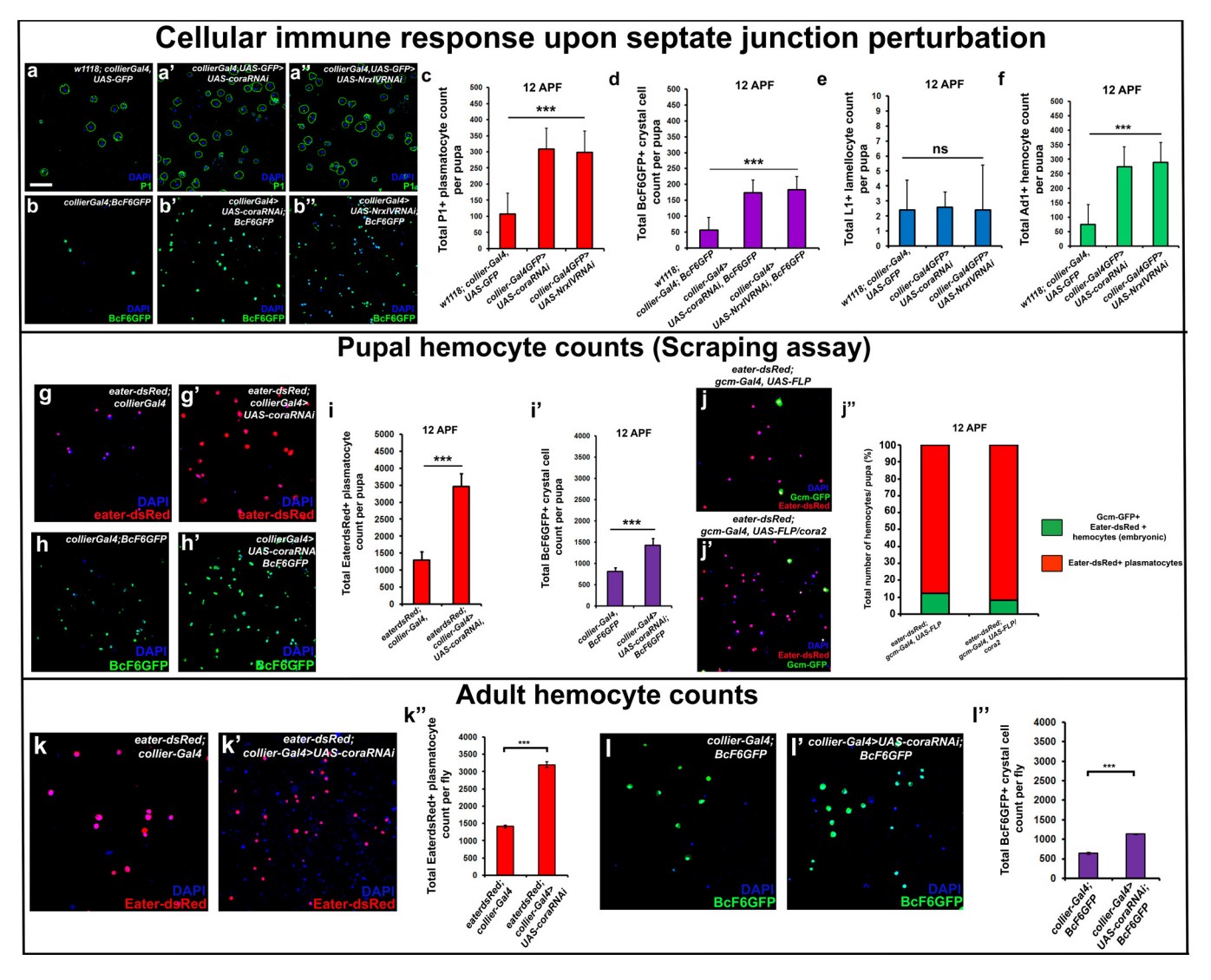

**Figure 6.** Depletion of septate junctions in the PSC triggers the hemocyte-mediated cellular immune response. (a–b'', c, d, f) Total number of P1 (Green) positive circulating plasmatocytes, BcF6GFP (Green) positive circulating crystal cells and Ad1 positive circulating adult hemocytes is highly increased in 12 hr APF (after puparium formation) early pupal circulation upon *collier-Gal4* mediated depletion of Cora or NrxIV as compared to the controls. (e) L1 positive circulating lamellocyte counts are not affected. (g–i') Eater-dsRed (Red) positive plasmatocyte or BcF6GFP (Green) positive crystal cell counts in 12 hr APF pupae using the scraping method for hemocyte isolation upon *collier-Gal4* mediated PSC- specific depletion of Coracle as compared to its control. (j–j'') Total number of hemocytes (represented as %) in 12 hr APF pupae using hemocyte scraping assay in pupae where the *Act5C > FRT > CD2>FRT > Gal4* and *UAS-FLP* and *UAS-GFP* (Green) transgenes were used to permanently label the hemocytes that express the embryonic plasmatocyte driver, *gcm-Gal4* in wild type and cora mutant genetic background with Eater-dsRed (Red) as the plasmatocyte marker. (k–k'') EaterdsRed (Red) positive adult plasmatocytes isolated from adult flies are increased upon PSC- specific depletion of Coracle using *collier-Gal4* as compared to the control. (l–l'') BcF6GFP (Green) positive adult crystal cells isolated from adult flies are also increased upon PSC- specific depletion of Coracle using *collier-Gal4* as compared to its control. Nuclei are labeled with DAPI (Blue). *** indicates p<0.001, * indicates p<0.1 and ns indicates non-significant. Error bars represent s.e.m. Scale Bar: (a–b'', g–h',j–j', k–k' and l–l') 50 μm.

DOI: https://doi.org/10.7554/eLife.28081.032

The following figure supplement is available for figure 6:

**Figure supplement 1.** Depletion of septate junctions in the PSC does not trigger endogenous activation of the humoral immune response in the absence of bacterial infection.

DOI: https://doi.org/10.7554/eLife.28081.033

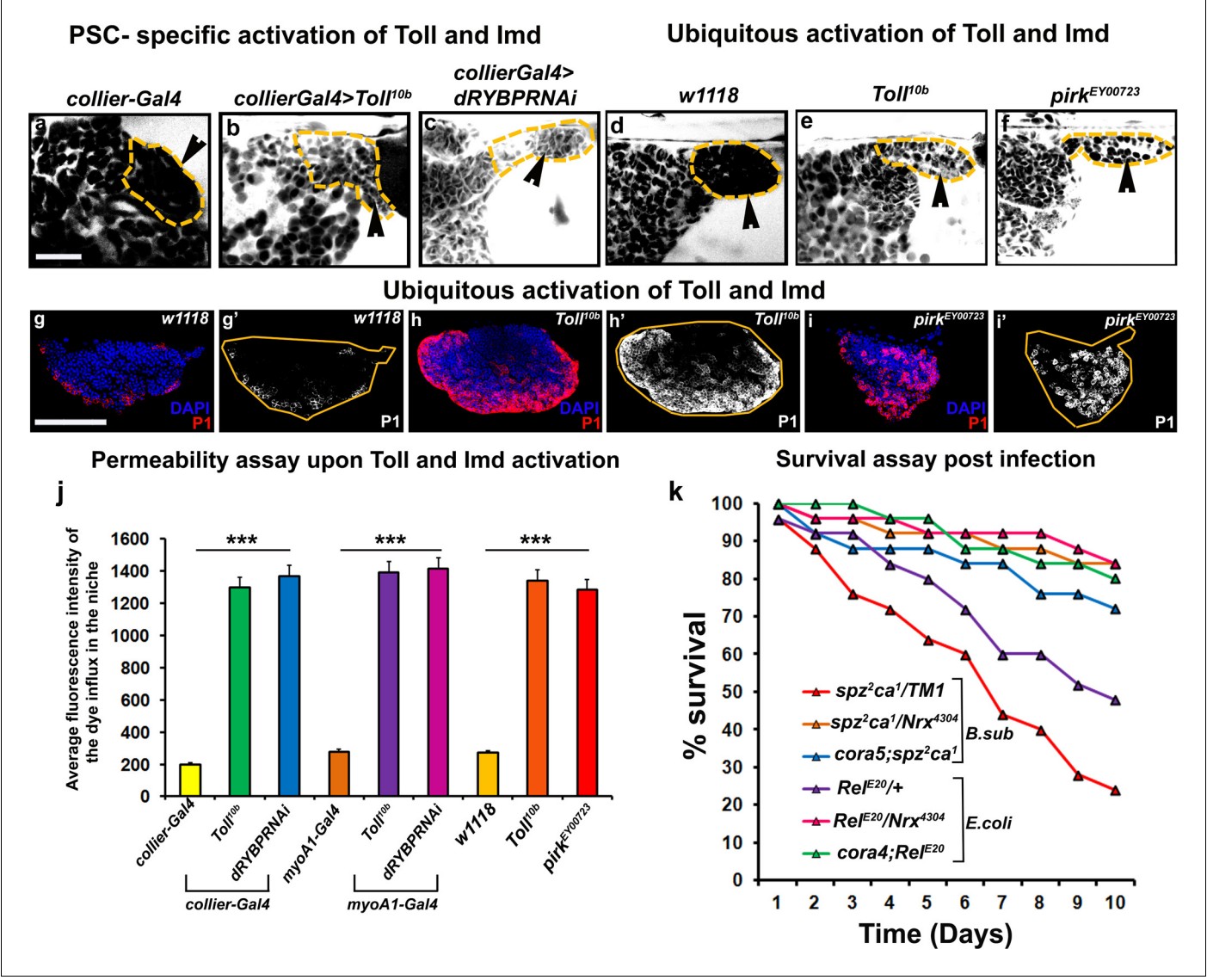

**Figure 7.** Barrier breakdown contributes to immune activation by triggering prohemocyte differentiation. (a–f) 70kDa-dextran (white) is excluded from the PSC in controls (a,d) but labels the PSC in genetic backgrounds that activate the Toll (b,e) or Imd (c,f) pathways either in the PSC specifically (b,c) or ubiquitously (e,f); quantitation of this data (j). (g–i) Ubiquitous activation of the Toll (h) and Imd (i) pathways results in plasmatocyte differentiation compared to wild-type controls (g). (k) Survival Plots post infection with *B. subtilis* or *E. coli* of Toll (*spz²ca¹*) and Imd (*Rel^{E20}*) pathway mutants combined with hypomorphic *NrxIV* and *cora* alleles. (b,c) (l–q') PSC labelled with GFP (green; *collier-Gal4 >UAS* GFP). Plasmatocytes labeled with P1 (Red:g-i,l-q; white:g'-i',l'-q'). Nuclei labeled with DAPI (Blue). ***=P < 0.001. Error bars represent s.e.m. Scale Bar:(g–i', l–q') 50 μm,(a–f) 40 μm.

DOI: https://doi.org/10.7554/eLife.28081.034

The following source data and figure supplements are available for figure 7:

**Source data 1.** Contains numerical data for quantitation in *Figure 7j*.

DOI: https://doi.org/10.7554/eLife.28081.037

**Figure supplement 1.** Toll and Imd pathway components are expressed in the lymph gland and regulate lymph gland mediated cellular immune response to infection.

DOI: https://doi.org/10.7554/eLife.28081.035

**Figure supplement 2.** Toll and Imd pathway activation alter the homeostatic balance in the lymph gland.

DOI: https://doi.org/10.7554/eLife.28081.036

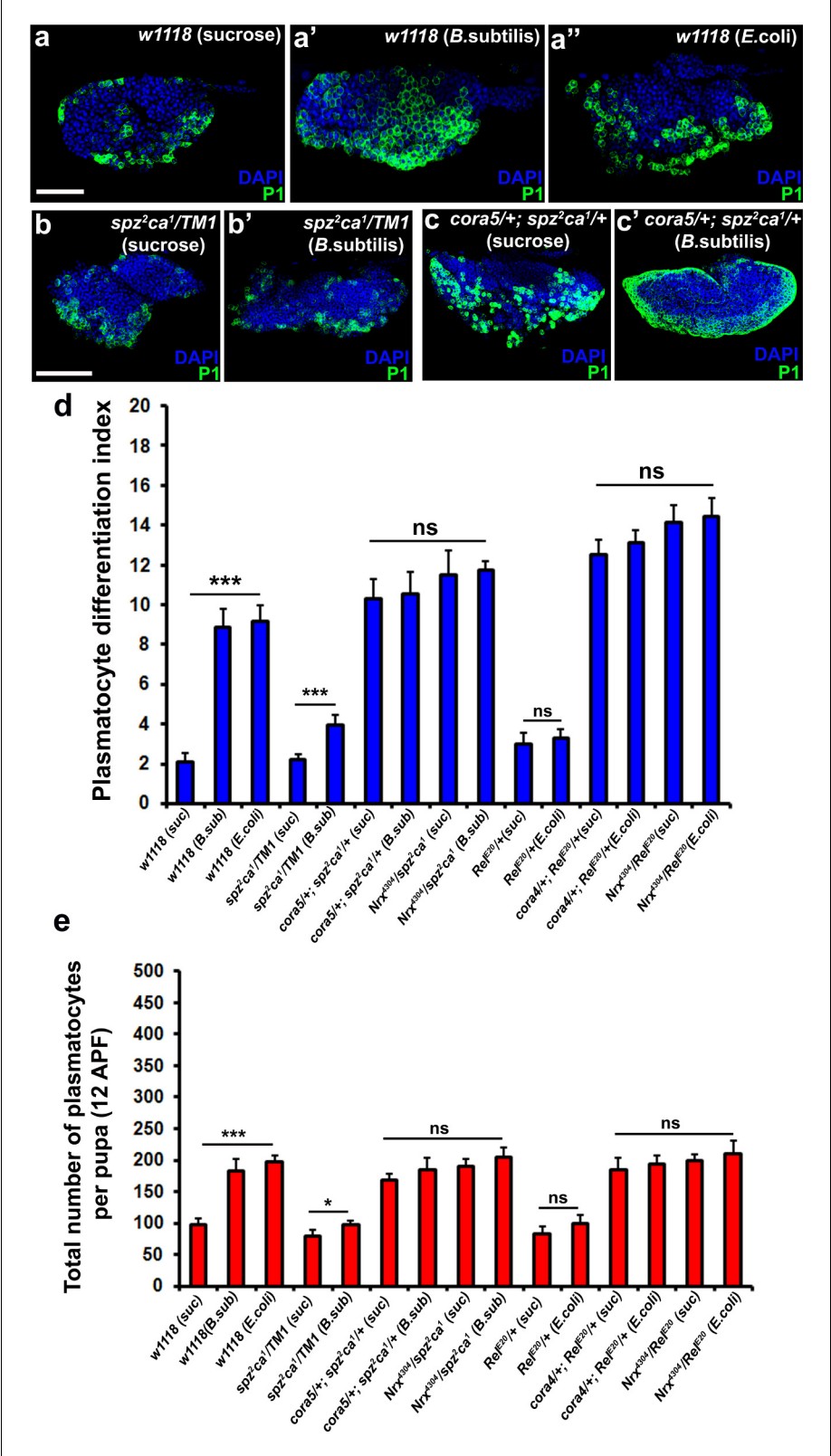

**Figure 8.** Modulation of permeability barrier at the PSC is a key target of both Toll and Imd pathway in order to mount a robust immune response. Number of P1 (Green) positive plasmatocytes (b–b') in $spz^2ca^1$ immune mutants is lower in uninfected or infected conditions as compared to respective wild type controls (a–a''). P1 (Green) positive differentiated plasmatocyte (c–c') numbers are increased in double mutants of $spz^2ca^1$ with *cora* mutant allele in both uninfected and infected conditions as compared to respective wild type controls. (d) Quantitation of plasmatocyte differentiation index in

*Figure 8 continued on next page*

*Figure 8 continued*

the immune deficient mutants and immune deficient mutants combined with mutations in SJ components in uninfected and infected conditions. (**e**) Total number of circulating plasmatocytes in *spz²ca¹* or *Relᴱ²⁰* immune mutants is lesser in uninfected or infected conditions as compared to respective wild type controls. Total number of circulating plasmatocytes is highly increased in double mutants of *spz²ca¹* with cora or nrxIV mutant allele or *Relᴱ²⁰* with *cora* or *nrxIV* mutant allele in both uninfected and infected conditions as compared to respective wild type controls. Nuclei are labeled with DAPI (Blue). *** indicates p<0.001, * indicates p<0.1 and ns indicates non-significant. Error bars represent s.e.m. Scale Bar: (**a–a''**) 50 µm (**b–c'**) 60 µm.

DOI: https://doi.org/10.7554/eLife.28081.038

The following figure supplement is available for figure 8:

**Figure supplement 1.** Cellular immune response is upregulated upon partial loss of septate junctions in the immune-deficient mutants.

DOI: https://doi.org/10.7554/eLife.28081.039

freely. For example, in such a model, occluding junctions could facilitate the build-up of high concentrations of signaling molecules inside the barrier. Data consistent with such a mechanism is derived from experiments looking at Wingless signalling. Specifically, the levels of the Wg ligand at the PSC were reduced by depletion of SJ components in the PSC but increased by overexpression of SJ components in the PSC as compared to the controls (***Figure 9h*** and ***Figure 9—figure supplement 1m–p''***). Upon depletion of SJ components Wg levels were observed to be lower throughout the LG, and not just the PSC, consistent with the reduction in number of prohemocytes which have been shown to express Wg (***Sinenko et al., 2009***; ***Figure 9—figure supplement 1m–p''***). Also in line with this model, we find that the expression of dMyc, which is typically downregulated by BMP signals in the PSC, is elevated following SJ knockdown in the PSC suggesting a reduced level of BMP signal (***Figure 9—figure supplement 2d–f',h***; ***Pennetier et al., 2012***). Intriguingly, elevating the levels of the Wg ligand at the PSC rescued prohemocyte differentiation phenotypes caused by Cora or NrxIV knockdown (***Figure 9a–g*** and ***Figure 9—figure supplement 1a–l',q***). Similarly depletion of dMyc in the PSC following knockdown of SJ components rescues their PSC cell number phenotype though it does not affect plasmatocyte differentiation (***Figure 9—figure supplement 2i–l', o,m–n'***). In addition to changes in the diffusion of PSC derived signals, depletion of SJ may modify the PSC microenvironment in other ways. For example the expression of Dally like protein (Dlp) was decreased following SJ knockdown in the PSC (***Figure 9—figure supplement 2a–c',g***). Dlp, a heparan sulfate proteo-glycan, is highly expressed in the PSC, where it regulates BMP signaling (***Pennetier et al., 2012***). The observed decrease in Dlp following SJ depletion can account for some of the phenotypes we observe such as an increase in the number of PSC cells (***Figure 2a–d***, ***Figure 2—figure supplement 3a''–c''',g',h–j',n*** and ***Figure 2—figure supplement 4d–f',n***; ***Pennetier et al., 2012***). Of note, the relation between SJ components and the Wingless and BMP signaling pathways is not bi-directional since the expression of Cora is not altered by abrogation of either pathway (***Figure 9—figure supplement 3a–d***). These data illustrate how changes in the PSC signaling microenvironment impact the regulation of hemocyte differentiation.

## Discussion

Our work here provides a novel mechanism for the induction of the cellular arm of the immune response upon infection. Specifically, our data suggests that increased differentiation of immune cells post-infection is mediated via the breakdown of a permeability barrier around the hematopoietic niche. Loss of the permeability barrier at the niche modifies the signalling microenvironment in such a way as to drive blood progenitors towards differentiation. As a result of this sequence of events there is increased availability of hemocytes in the circulating hemolymph to fight infection. Intriguingly, we find that inducing hemocyte differentiation by breaking down the permeability barrier in larva has long lasting effects including an increase in the number of circulating hemocytes in adults, increased pathogen clearance, and substantial improvement in the ability of adult flies to fight infection. This effect is specific to the PSC as depleting SJ components in other tissues and organs did not result in any changes in the LG. Importantly, we find that breakdown of the permeability barrier is both necessary and, at least in part, sufficient for inducing the cellular arm of the immune response. Taken together, our results argue that the novel mechanism we uncovered, involving the regulation of SJ at the PSC, is an important and powerful component of the regulatory network that controls the innate immune response in flies.

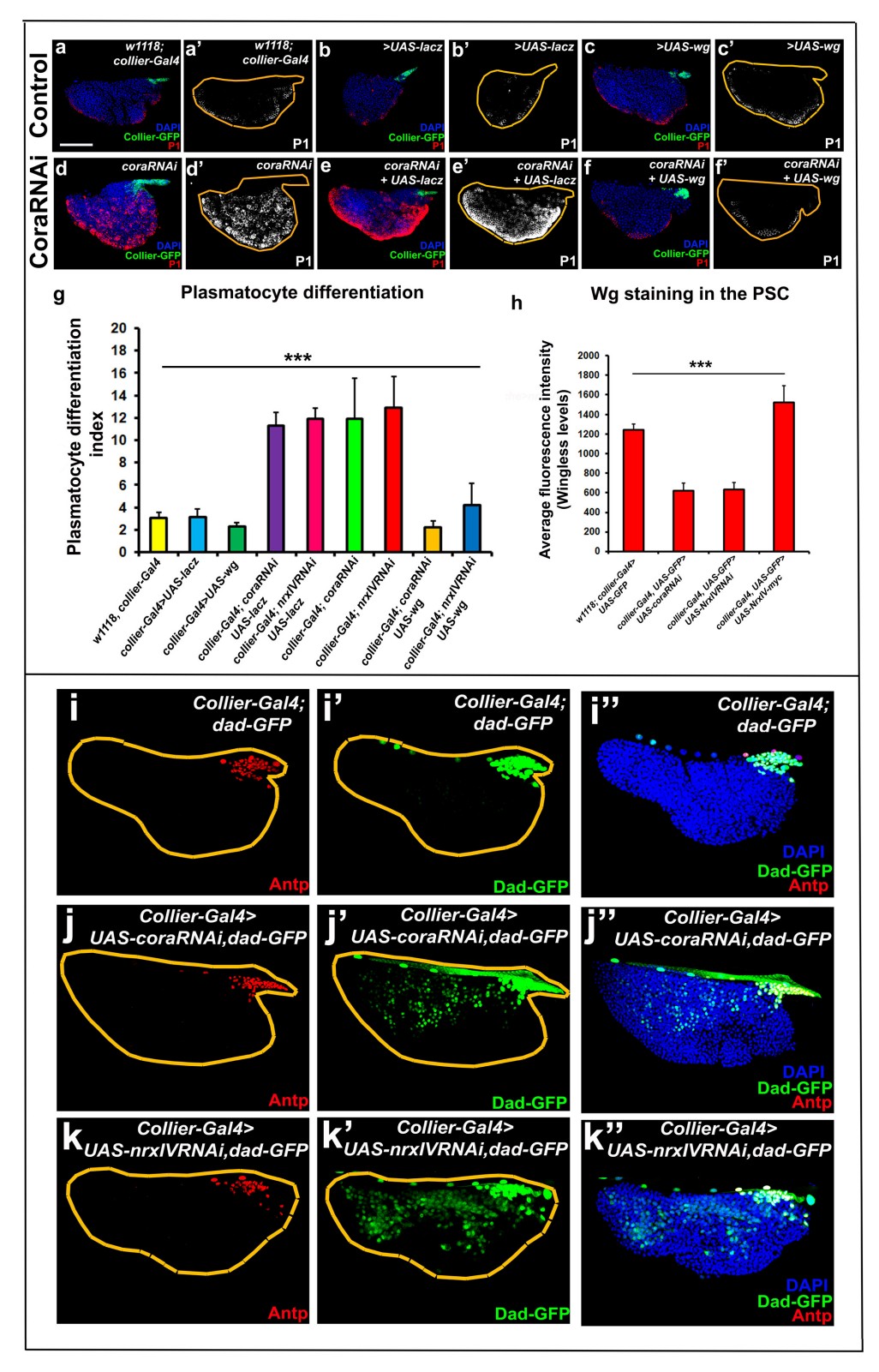

**Figure 9.** Signalling microenvironment at the PSC is altered upon PSC-specific depletion of septate junction components. (**a–c'**) Plasmatocyte differentiation in controls: *collier-Gal4, collier-Gal4 >UAS LacZ, collier-Gal4 >UAS* wg. (**d,d'**) Cora depletion (*collier-Gal4 >UAS* coraRNAi) increases plasmatocyte differentiation but this defect is rescued by co-expression of wg (**f,f'**, *collier-Gal4 >UAS coraRNAi, UAS-wg*) but not LacZ (**e,e'**, *collier-Gal4 >UAS coraRNAi, UAS-LacZ*). (**g**) Quantitation of prohemocyte differentiation into plasmatocytes upon wg over-expression in Nrx or Cora depleted

*Figure 9 continued on next page*

*Figure 9 continued*

PSC. (**h**) Quantitation of Wg levels in the PSC analyzed by immuno-staining with anti-Wg antibody. Compared to control, lower staining is observed following *collier-Gal4* mediated *cora* or *NrxIV* depletion in the PSC whereas higher levels are observed after over-expression of *UAS-NrxIV-myc* in the PSC. Range of BMP signals marked by Dad-GFP (Green) is extended further from the PSC upon *collier-Gal4* mediated depletion of *cora* (**j–j''**) or *NrxIV* (**k–k''**) as compared to the control (**i–i''**). (**a–f'** and **i–k''**) PSC labelled with GFP (green; *collier-Gal4 >UAS* GFP) or with Antennapedia (Red:**i-k**). Plasmatocytes labeled with P1 (Red:**a–f**; white:**a'–f'**). Nuclei labeled with DAPI (Blue). ***=$P < 0.001$. Plasmatocyte differentiation indices of lymph glands bearing PSC- specific lacZ or Wg over-expression in the PSC following *cora* or *NrxIV* knockdown were compared to respective parental controls or *cora/NrxIV* knockdown genotypes for the statistical analysis with the t-test. Error bars for plasmatocyte differentiation data represent s.e.m. Error bars for the wg fluorescence intensity data represent s.d. Scale Bar:(**a–f'**, **i–k''**) 50 μm.

DOI: https://doi.org/10.7554/eLife.28081.040

The following source data and figure supplements are available for figure 9:

**Source data 1.** Contains numerical data for quantitation in *Figure 9g*.
DOI: https://doi.org/10.7554/eLife.28081.044
**Source data 2.** Contains numerical data for quantitation in *Figure 9h*.
DOI: https://doi.org/10.7554/eLife.28081.045
**Figure supplement 1.** Wg over-expression suppresses prohemocyte differentiation induced by depletion of septate junction components in the PSC.
DOI: https://doi.org/10.7554/eLife.28081.041
**Figure supplement 2.** Depletion of septate junction in the PSC changes the signalling micro-environment of the PSC affecting PSC cell numbers and blood cell homeostasis in the LG.
DOI: https://doi.org/10.7554/eLife.28081.042
**Figure supplement 3.** Abrogation of Wg or BMP signalling in the PSC does not affect Coracle expression in the PSC.
DOI: https://doi.org/10.7554/eLife.28081.043

Our current study focused on the role of SJ components specifically in the primary lobe of the LG. It is unclear whether SJs are also involved in regulating hematopoiesis in the posterior lobes (*Jung et al., 2005*; *Kalamarz et al., 2012*; *Letourneau et al., 2016*). Currently there is a great deal more known about the regulation and function of the primary lobe of the LG, but as we learn more about the posterior lobes it will be worthwhile to investigate the possible role of SJ in the posterior lobes. By focusing on the primary lobes we were able to analyze the fate of prohemocytes following SJ depletion and observed a dramatic reduction in their number concomitant with hemocyte differentiation. These results suggest one possible function of the permeability barrier around the PSC is to assist in prohemocyte maintenance, perhaps by altering the signaling environment around the prohemocytes.

How would changes in the SJ-mediated permeability barrier around the PSC affect the signalling microenvironment? Molecules can cross tight-junction-mediated permeability barriers via two mechanisms: small (4 Å) selective pores or larger non-selective pores that may represent gaps in the barrier (*Shen et al., 2011*). The 40–70 kDa exclusion size observed at the PSC is consistent with the large-pore non-selective mechanism and is larger than that seen to date for epithelia in the fly, which is in the range <10 kDa, perhaps indicating the LG SJs are highly specialized (*Lamb et al., 1998*; *Paul et al., 2003*; *Genova and Fehon, 2003*). Secreted signalling proteins known to regulate prohemocyte differentiation such as Wg and Dpp, are within the 40–70 kDa exclusion size range. Infection can therefore change the signalling microenvironment at the PSC by down-regulating occluding junction components allowing signals to escape the niche. Consistent with this hypothesis our analysis of Dpp and Wg signalling illustrate how changes in occluding junctions can affect the concentration of signalling molecules both inside and possibly outside the PSC. These changes in concentration can manifest in diverse ways as a permeability barrier can both concentrate signals, as appears to be the case for Wg, or alternatively restrict the range and concentration of signals, as is the case for Dpp. For example, we are able to rescue the prohemocyte differentiation defect caused by depletion of SJs by restoring Wg levels in the PSC. This rescue might be due to effects of Wg on the niche itself, as Wg is required to maintain the niche, or by restoring a steep concentration gradient of Wg that SJs help in creating. A second example is provided by dMyc, which is usually repressed in the PSC by high levels of BMP signal, is de-repressed as a result of SJ depletion and the subsequent drop in BMP levels in the PSC. Depleting dMyc in the PSC following SJ component knockdown substantially rescue the phenotype caused by loss of the permeability barrier. Alternatively, SJs can shape the signaling environment at the niche in other ways as suggested by the

altered distribution of Dlp upon knockdown of SJs, a result which hints at the kind of additional, more direct, roles for occluding-junction components in mediating signal transduction (*Matter and Balda, 2003*). Full understanding of the effects mediated by modulation of occluding junctions on the PSC microenvironment will require accounting for changes in all possible signalling molecules present in the niche; this will be the focus of future studies.

The breakdown of the permeability barrier in the PSC may represent an evolutionary adaptation to infection, since pathogens are known to attack such barriers as this provides increased access to the tissues in the body (*Guttman et al., 2006*; *Guttman and Finlay, 2009*). Indeed, in vertebrates, activation of the immune system and interferon production are also known to cause a reduction in the expression of occluding junction components and breakdown of permeability barriers (*Watson et al., 2005*; *Edelblum and Turner, 2009*). It is tempting to speculate that as pathogens developed the ability to induce breakdown of permeability barriers animals adapted by using the degradation of such barriers as a cue to activate the cellular immune response.

As evidenced by our previous studies in the testes (*Fairchild et al., 2015*; *Fairchild et al., 2016*), modulation of occluding junctions may represent a general feature of stem-cell niche systems. Consistent with this hypothesis, prominent expression of tight junction and/or the presence of a permeability barrier has already been described as integral to many vertebrate stem cell niche models including the adult neural stem cells (*Tavazoie et al., 2008*), germline stem cells (*Oatley and Brinster, 2012*) and human hair follicles (*Brandner et al., 2003*; *Meyer-Blazejewska et al., 2011*). Taken together, our work suggests that occluding junctions play a novel and important role in shaping the stem cell microenvironment to regulate stem cell maintenance and differentiation.

## Materials and methods

### Fly stocks

*w1118* was the wild type control strain used. PSC drivers used were *Pcol85Gal4* referred to as *collier-Gal4* driving *UAS-mCD8GFP* (gift from Michele Crozatier and Lucas Waltzer, Toulouse, France), *Ser9.6-Gal4* driving UAS-GFP (gift from Utpal Banerjee, UCLA) and Antp-Gal4 driving UAS-GFP (gift from Lucas Waltzer). Prohemocyte (medullary zone) specific driver used was *tep4-Gal4* driving *UAS-mCD8GFP* or *dome-Gal4* driving *UAS-mCD8GFP* (gift from Lucas Waltzer, Toulouse, France), *gcm-Gal4* (kind gift from Lucas Waltzer), *Dad-GFP* (BMP signalling reporter, gift from Douglas Allan, Vancouver), *UAS-NrxIV-myc* (gift from Manzoor Bhat), *UAS-myc-cora$^{1-1698}$* (gift from Richard Fehon, Chicago), *Drosomycin-GFP* (Toll pathway reporter, gift from Dominique Ferrandon), *Diptericin-GFP* (Imd pathway reporter, gift from Dominique Ferrandon), *eater-dsRed* (kind gift from Dr. Elio Sucena) *BcF6GFP* and *BcF6mcherry* (gift from Robert Schulz, Notre Dame). Gut specific driver *myo1AGal4* (gift from Leanne Jones, UCLA). RNAi lines for NrxIV (JF03142, TRiP, RRID:BDSC_28715), Cora (HM05144, TRiP, RRID:BDSC_28933), additional RNAi lines for Cora (VDRC, GD1405, RRID:BDSC_28933), NrxIV (VDRC, GD2436), dRYBP (HMS00931, TRiP, RRID:BDSC_33974), constitutively activated Toll mutant (*Toll$^{10b}$*, RRID:BDSC_30914), Imd pathway mutant (*pirk$^{EY00723}$*,RRID:BDSC_15039), *UAS-Toll$^{10b}$* (RRID:BDSC_58987), *UAS-wg*, *UAS-lacZ*, all the septate junction and associated mutant alleles *cora2/CyO*, *cora4/CyO*(RRID:BDSC_52232), *cora5/CyO* (RRID:BDSC_52233), *cora14/CyO* (RRID:BDSC_9099), *Nrx$^{4304}$/TM6B* (RRID:BDSC_4380), *ATPα$^{DTS1R2}$/+*, *sinu$^{nwu7}$/TM6B* (RRID:BDSC_8577), *kune$^{C309}$/CyO* (RRID:BDSC_16333), *Mcr$^{EY07421}$/CyO* (RRID:BDSC_15997), pan-neuronal driver *elav-Gal4*, fat body specific *lsp2-Gal4*, *w1118* and *oregon-R* (were used as wild-type controls), Canton-S, Spaetzle mutant (*spz$^2$ca$^1$*/TM1, RRID:BDSC_3115), Relish mutant (*Rel$^{E20}$*, RRID:BDSC_55714), *UAS-GFP*, *UAS-mCD8GFP*, *AB1-Gal4* (RRID:BDSC_1824), *Mef2-Gal4*, *Act5C > FRT > CD2>FRT > Gal4*, *UAS-FLP* (all were obtained from Bloomington Stock Center, USA), Neurexin-IV::GFP from FlyTrap (CA06597) (*Buszczak et al., 2007*), *ATPα$^{CPTI002761}$* (gift from Dr. Vanessa Auld, UBC), *UAS-dMycRNAi* (gift from Dr. Savraj Grewal, University of Calgary), *UAS-tkvDN* (Dominant negative) were obtained from Dr. Douglas Allan, UBC, *UAS-Dfz2ECD-GPI* (Bloomington, RRID:BDSC_44221), *dome-MESOlacZ* reporter (originally line made by Martin Zeidler, University of Sheffield, flies gifted by Dr. Nancy Fossett)

### *Drosophila* genetics

PSC driver used was either *Pcol85Gal4* (referred to as *collier-Gal4*) driving *UASmCD8GFP*, *Antp-Gal4* driving *UAS-GFP* or *Ser9.6-Gal4* driving *UAS-GFP*. Medullary zone driver used was *tep4-Gal4* driving *UASmCD8GFP* or *dome-Gal4* driving *UAS-mCD8GFP*. Niche specific knockdown of Nrx and Cora was achieved by using *UAS-NrxIVRNAi* or *UAS-coraRNAi* driven by *Pcol85Gal4*. Systemic knockdown of SJ's was done using gut specific *myo1A-Gal4*, fat body specific *lsp2-Gal4*, muscle specific *mef2-Gal4*, salivary gland specific *AB1-Gal4* or pan-neuronal *elav-Gal4* knockdown of SJ's. Toll and Imd activation in the PSC and gut was achieved by expressing *UAS-Toll10b* or *UAS-dRYBPRNAi* using *Pcol85Gal4* or *myo1AGal4* respectively. *Toll10b* and *pirk$^{EY00723}$* mutants were used to assess ubiquitous activation of Toll and Imd pathways. Flies bearing *UAS-wg* transgene were used for over-expression of Wingless in niche specific Cora or Nrx depleted flies using *Pcol85Gal4*. *UAS-lacZ* transgenic construct was used as a control for the rescue experiments with Wg where *UAS-lacZ* was expressed in the genetic background of PSC specific Cora or NrxIV knockdown. Cora knockdown in the PSC was validated using anti-Coracle antibody. For analysis of BMP reporter activity upon SJ perturbation in the PSC, *collier-Gal4* mediated Cora or NrxIV knockdown was done in *Dad-GFP* (BMP reporter) genetic background to assess the status and range of BMP signalling. Mutant alleles of *cora* were recombined to obtain allelic combinations of *cora2/cora14* and *cora5/cora14* in cis-heterozygous combinations which were further used for lymph gland analysis. *spz$^2$ca$^1$* and *Rel$^{E20}$* mutants were recombined with mutant alleles of Nrx (*Nrx$^{4304}$/TM6B*) and Cora (*cora4/CyO*, *cora5/CyO*) to obtain trans-heterozygotes that were compared to *spz$^2$ca$^1$/TM1* and *Rel$^{E20}$/+* respectively and used for survival analysis upon infection. These double mutants were also used to analyze various lymph gland parameters like PSC, plasmatocyte, crystal cell numbers and to assay for the circulating hemocyte numbers in the early pupal stages. SJ over-expression experiments were performed using *UAS-myc-cora$^{1-1698}$* or *UAS-NrxIV-myc* lines that were either driven by PSC, MZ or CZ specific drivers. PSC specific over-expression of the myc-tagged Cora or NrxIV constructs was confirmed using anti-myc antibody. PSC specific over-expression of Cora or NrxIV using *UAS-myc-cora$^{1-1698}$* and *UAS-NrxIV-myc* constructs driven by *collier-Gal4* was done in the genetic background of *cora* or *NrxIV* mutants to analyze for rescue of the SJ mutants. Gut specific over-expression of Cora using *UAS-myc-cora$^{1-1698}$* construct driven by *myo1A-Gal4* was done in the *cora* mutant genetic background to analyze for rescue. Humoral immune response was assessed using *Drosomycin-GFP* or *Diptericin-GFP* reporters for assessing Toll and Imd activation respectively in the fat body upon depletion of SJ using PSC specific *collier-Gal4* in uninfected and infected conditions. Toll expression in the PSC was also validated by using PSC specific knockdown of Toll and then detecting the down-regulation of Toll expression using anti-Toll antibody. Analysis of circulating hemocytes was done upon knockdown of septate junctions using PSC specific *collier-Gal4*. Plasmatocytes, lamellocytes and adult hemocytes were identified by P1, L1 and Ad1 antibodies respectively while circulating crystal cells were identified by C5 antibody or by using *BcF6GFP* or *BcF6mcherry* transgenes in the genetic background of septate junction knockdown in the PSC. Flies carrying the embryonic plasmatocyte-specific driver *gcm-Gal4* recombined with a *UAS-FLP* were crossed to a strain bearing a flip-out cassette (*Act5C > FRT > CD2>FRT > Gal4*) and a *UAS-GFP*. These flies were then crossed to eater-dsRed flies to track plasmatocytes which were analyzed in wild type or cora mutant background. To analyze plasmatocyte differentiation using an alternate plasmatocyte marker - Eater-dsRed (tagged line to mark plasmatocytes), *eater-dsRed* strain was placed in the genetic background of *collier-Gal4* mediated PSC specific depletion of Coracle and compared to the wild type control genetic background. For analyzing Eater-dsRed expression upon systemic infection, flies bearing the eater-dsRed transgene were crossed to *collier-Gal4* flies expressing *UAS-GFP* in the PSC. For investigating the total number and proportion of prohemocytes in the PSC-specific SJ depletion background, the transgenic reporter, *Dome-MESOlacZ* was used which marks the prohemocytes. This reporter was placed in the genetic background of *collier-Gal4* mediated SJ depletion. For the mechanistic analysis of BMP and Wg signalling with dMyc, dMyc was depleted using RNAi in the genetic background of *collier-Gal4* mediated SJ depletion. For inactivation of Wg or BMP signalling in the PSC, a dominant negative form of wingless receptor, Frizzled2 (*UAS-dfz2ECD-GPI*) or BMP receptor, Thickvein (*UAS-tkvDN*) was expressed in the PSC using *collier-Gal4*. Third instar larval progeny was then used for systemic pricking and further analysis. For all the knockdown and over-expression experiments, the respective Gal4 drivers were outcrossed to *w1118* and those were then

used as respective controls. All the lymph gland analysis was done on wandering third instar larval lymph glands. All the fly stocks and genetic crosses were maintained at 25°C.

## Antibodies used

Unless otherwise indicated all antibodies were obtained from the Developmental Studies Hybridoma Bank, Iowa, USA. Rabbit anti-GFP (1:500, A11122, Molecular Probes, A11122 RRID:AB_221569), mouse anti-Antennapedia (1:50, 8C11, DSHB Cat# anti-Antp 8C11 RRID:AB_528083), mouse anti-Hindsight (1:50, 1G9, DSHB-GFP-1G9 RRID:AB_2617420), mouse anti-Coracle (1:250, C566.9 (RRID: AB_1161642) and C615.16 (RRID:AB_1161644)), mouse anti Wg (1:50, 4D4, RRID:AB_528512), mouse anti Relish (1:25, 21F3, RRID:AB_1553772), rabbit anti Toll (1:50, d-300/sc33741, Santa Cruz Biotechnology, Inc., RRID:AB_672892), mouse anti-P1 (1:100, NimRodC1), mouse anti-L1 (1:25), mouse anti Ad1 (1:2), mouse anti-C5 (1:10) (gift from Dr. Istvan Ando, Szeged, Hungary), anti-myc antibody (1:500, gift from Dr. Shernaz Bamji, UBC, Vancouver), Rabbit anti-dMyc (1:250, gift from Dr. David Stein, University of Texas at Austin), mouse anti-Dlp (1:50, 13G8, DSHB, RRID:AB_528191), Rabbit anti-Sinuous and anti-Kune-kune (1:500, gift from Dr. Greg Beitel, Northwestern University), Rabbit anti-betagalactosidase (1:1000, gift from Dr. Douglas Allan, UBC)

## Other reagents used

10 kDa dextran conjugated to Alexa Fluor 647 (D-22914, Molecular Probes), 40 kDa dextran conjugated to Tetramethylrhodamine (D-1842, Molecular Probes) and 70 kDa dextran conjugated to Rhodamine B (D-1841, Molecular Probes), for the Dual Dye Assay – 70 kDa dextran conjugated to Oregon green (D7176, ThermoFisher Scientific) and 40 kDa dextran conjugated to Tetramethylrhodamine (D-1842, ThermoFisher Scientific), Alexa Fluor – 488 Phalloidin (1:200, A12379, ThermoFisher Scientific, A-12379 RRID:AB_2315147), VECTASHIELD with DAPI (H-1200, Vector Laboratories, RRID:AB_2336790).

## Lymph gland dissection and immunohistochemistry

Wandering third instar larvae were used for the dissection of lymph gland. The dissections were done in Phosphate Buffer Saline (PBS), fixed in 4% paraformaldehyde (PF), followed by washes with 0.1% PTX (PBS with 0.1% Triton-X). The lymph gland preparations were then blocked with 1% Normal Goat Serum (ab7481, abcam) followed by overnight primary antibody incubation at 4°C. The primary antibody incubation was followed by washes with 0.1% PTX and block treatment. Appropriate Alexa-Fluor–conjugated secondary antibodies were used. The lymph gland preparations were incubated in the secondary antibody for 2 hr at room temperature followed by three washes with 0.1% PTX and then mounted in VECTASHIELD with DAPI (H-1200, Vector Laboratories). All the antibody dilutions were made in PBS.

## Dye permeability assay

Larval lymph glands were dissected in Schneider's Drosophila Medium (Catalog number: 21720–024, Gibco). Lymph glands were transferred to medium containing dye (10 kDa, 40 kDa or 70 kDa dextran at a final concentration of 0.2 µg/µl). Lymph glands were imaged within 60 min of dye addition. Images were acquired from near the imaging surface to minimize out-of-plane fluorescence from dye in the medium. Comparable detection thresholds were ensured by setting the exposure level in the medium outside the primary lymph gland lobe to saturation level for image acquisition. PSC region was identified using the GFP driven by either *collier-Gal4* or Antp-Gal4.

## Dual Dye Assay (DDA)

Larval lymph glands were dissected in Schneider's Drosophila Medium (Catalog number: 21720–024, Gibco). Lymph glands were then transferred to medium containing Oregon green conjugated 70 kDa dye (D7176, ThermoFisher Scientific) and Tetramethylrhodamine conjugated 40 kDa dye (D1842, ThermoFisher Scientific at a final concentration of 0.2 µg/µl). Lymph glands were imaged within 60 min of dye addition. Images were acquired from near the imaging surface to minimize out-of-plane fluorescence from dye in the medium. Comparable detection thresholds were ensured by setting the exposure level in the medium outside the primary lymph gland lobe to saturation level

for both the dyes for image acquisition. PSC region was identified as the region just anterior to the first pair of pericardial cells which lie immediately posterior to the PSC cells.

## Image acquisition and analysis

All images were acquired on an Olympus FV1000 inverted confocal microscope. Image analysis was performed in Olympus Fluoview (Ver.1.7c) and ImageJ software. Lymph gland boundaries have been indicated with white or brown lines. White or black arrowheads have been used to indicate the inter-mingling of a subset of differentiated hemocytes with the PSC cells or to indicate the influx of dye upon SJ perturbation, Toll or Imd activation or bacterial infection. PSC has been marked with white or yellow borders in the images for the dye permeability assay. Nuclei are stained with DAPI. Images were processed using Adobe Photoshop CS3. Images were processed uniformly for brightness and contrast using Adobe Photoshop CS3 wherever needed.

## Quantitation of dye permeability assay

Fluorescence intensity profile of the dye influx in the niche was obtained using the ImageJ software (NIH) by selectively placing the ROI (Region of interest) tool between two given cells in the HSC niche region. Average fluorescence intensity of the dye influx in the niche was then used for plotting the graphs. For other lymph gland hemocytes, ROI tool was drawn around the hemocytes in the primary lymph gland lobe to obtain fluorescence intensity values.

## Quantitation of co-localization of the two dyes for the Dual Dye Assay (DDA)

Pearson's co-localization co-efficient was obtained using Co-localization Finder (ImageJ plugin, NIH) by selectively placing the ROI (Region of Interest) tool around the cells in the PSC or around the other hemocytes (non-PSC cells) in the primary lymph gland lobe.

## PSC cell counts

Antennapedia positive cells were counted using cell counter plugin in ImageJ software and the average number of antennapedia positive cells were plotted.

## Electron microscopy

Lymph glands were dissected in fixative [1.5% paraformaldehyde (Sigma-Aldrich, Missouri, USA), 0.1 M sodium cacodylate (Electron Microscopy Sciences), 1.5% glutaraldehyde (Electron Microscopy Sciences), adjusted to pH 7, at room temperature. The lymph glands were fixed for 2–3 hr at room temperature. The samples were washed with 0.1 M sodium cacodylate (pH 7.3) (3 times for 10 min) and then post-fixed for 1 hr in 1% osmium tetroxide in 0.1M Na cacodylate on ice. The lymph glands were washed with $ddH_2O$ (3 times for 10 min), stained for 1 hr 'en bloc' with 1% aqueous uranyl acetate, again washed with $ddH_2O$ (3 times for 10 min) and then dehydrated through an ascending concentration series of ethyl alcohols ending with three changes (10 min each) of 100% ethyl alcohol. The dehydration was followed by two changes (15 min each) of 100% propylene oxide, and then the samples were placed in a 1:1 mixture of propylene oxide:EMBED 812 Resin (Electron Microscopy Sciences) overnight. The lymph glands were embedded in 100% EMBED 812 Resin and the resin polymerized for 48 hr at 60°C. Thin sections were cut on a Leica EM UC7 ultramicrotome (Leica Microsystems), picked up on 200 mesh copper grids (Electron Microscopy Sciences) and stained with uranyl acetate and lead citrate. Sections were viewed and imaged on a Tecnai G2 Spirit electron microscope (FEI North America NanoPort) operated at 120 kV. PSC region was specifically finely dissected from the primary lymph gland lobe by isolating the posterior most part of the primary lymph gland lobe which is located anterior to the first pair of pericardial cells. These cells were then used for EM acquisition and analysis.

## Estimation of prohemocyte differentiation indices using MatLab

To determine the number of cells in the lymph gland, we developed custom cell counting scripts in MatLab. This MatLab script has been uploaded as *Source code 1* along with a supporting accessory file, *Source code 2*. *Source code 1* was used for all the hemocyte quantitation analysis. We first filtered every image in the z-stack in the DAPI channel using a difference of Gaussians approach. A

wide filter is used to remove background intensity, and a smaller filter is used to remove small objects. We applied each filter to the image and subtracted the result of the smaller filter from that of the wide filter, then thresholded the final image to generate a binary mask which effectively identified cell nuclei. The script then automatically identified the bright spots within the 3 dimensional image corresponding to nuclei in order to determine their numbers and centroid coordinates. To calculate differentiation index, we determined what proportion of these cells also expressed markers for differentiation. For each cell nucleus identified, we automatically measured intensity in the red channel within a search radius around the nucleus centroid. The search radius was defined as 1.5 times the average radius of nuclei. Nuclei surrounded by above threshold intensity in the red channel were considered differentiated. These numbers were then used to calculate the plasmatocyte and crystal cell differentiation indices. In addition to this, Dome-MESO lacZ reporter which is active in the medullary zone prohemocytes was used to estimate the total number of prohemocytes and the relative proportion of prohemocytes in the LG upon PSC- specific depletion of SJ components. Dome-MESO lacZ positive cells were identified using beta-Gal antibody and were counted using the MatLab script. This MatLab script was also used for estimating the circulating hemocyte counts.

## Larval oral infection assay

Early third instar larvae were first kept in empty vials and starved for 2–3 hr. Post starvation these larvae were placed in fly food vials covered with Whatman filter paper discs soaked with either 10% sucrose alone (control solution) or with concentrated bacterial pellets (*B. subtilis* or *E. coli*) suspended in 10% sucrose. 12 hr post infection, oral-infected larvae were used for the live dye permeability assay or lymph gland analysis post fixation.

## Adult oral infection and survival assay

For all the oral infection assays, freshly eclosed adult male flies (2–4 days old) were starved for 2–3 hr in empty vials at 29°C for synchronous bacterial feeding upon infection. The starved flies were flipped in fresh vials containing fly food media completely covered with Whatman filter paper disk soaked with either 10% Sucrose solution (control) or bacterial suspension in 10% sucrose (*B. subtilis* or *E. coli*) and kept at optimum temperature for bacterial growth. These vials (sucrose alone or bacterial suspension) were then flipped into fresh fly food vials 24 hr post incubation. These flies were flipped regularly and the survival is monitored over time (days). Number of male flies for each survival assay was n = 25. Survival assays were done in triplicates (N = 3) and data was pooled from these independent experiments.

## Larval systemic infection assay

Third instar larvae were washed three times with sterile ddH2O and pricked using a tungsten pin dipped in bacterial suspension of *B. subtilis* ($OD_{600}$ = 20) or *E. coli* ($OD_{600}$ = 200) on the postero-lateral part and then used for immunofluorescence analysis.

## Adult systemic infection assay

Freshly eclosed flies were pricked using a tungsten pin dipped in bacterial suspension of *B. subtilis* ($OD_{600}$ = 20) or *E. coli* ($OD_{600}$ = 200) on the thorax and were then used for survival assay.

## Survival assay upon systemic infection

Freshly eclosed flies were pricked using a tungsten pin dipped in bacterial suspension of *B. subtilis* ($OD_{600}$ = 20) or *E. coli* ($OD_{600}$ = 200) on the thorax. Sucrose solution was used as control. These vials (sucrose alone or bacterial suspension) were then transferred into fresh fly food and then flipped regularly to monitor survival over time (days). Number of male flies for each survival assay was n = 25. Survival assays were done in triplicates (N = 3) and data was pooled from these independent experiments.

## Estimation of in-vivo bacterial load

Flies of respective control and test genotypes were infected by pricking in the thorax with a sterile needle previously dipped in a concentrated pellet of Ampicillin resistant strain of *E. coli* (kind gift from Dr. Bret Finlay, UBC) grown at an $OD_{600}$ = 200. 24 hr post systemic infection flies were taken

individually and homogenized in 100 µl of sterile PBS, diluted serially and then plated onto LB-agar (Luria-Bertani) media plates supplemented with ampicillin (100 µg/ml). Plates were incubated overnight at 37° C. Bacterial colonies were then counted and the CFUs (Colony forming units) per fly were estimated and plotted. These were used as a measure of bacterial clearance ability of the flies. Results were averaged from three experimental repeats.

## Circulating hemocyte isolation and analysis (using integument incision method)

Circulating hemocytes were obtained by making a small incision in the pupal integument in early pupal stages (12 hr APF – after puparium formation) and the hemolymph was allowed to drain onto the coverslip and attach. Hemolymph was isolated in PBS and allowed to attach for 1 hr, followed by immunostaining. The incision made is a precise incision so as to prevent the tissue fluid and fat globules from oozing out of the pupal casing. Differentiated plasmatocytes were identified by P1 antibody, crystal cells were identified by *BcF6GFP* or *BcF6mcherry* transgenic construct or C5 antibody. Lamellocytes were identified by L1 antibody. Adult hemocytes were identified by Ad1 antibody. Circulating hemocytes were analyzed from *collier-Gal4* driven septate junction perturbed pupae. Hemocytes were counted using the MatLab software. Images were taken from ten different fields for each experiment (n = 10) using Olympus FV1000 microscope.

## Circulating hemocyte isolation and analysis (using scrape assay)

Circulating hemocytes were obtained from early pupae (12 hr APF) according to the hemocyte bleed/scrape assay as described (*Petraki et al., 2015*) and were subsequently counted using MatLab software. This method was also used to isolate hemocyte from adult flies to estimate the adult hemocyte counts.

## Circulating hemocyte counts using MatLab

To determine the number of circulating hemocytes, the custom cell counting script that was developed for estimating hemocyte differentiation in the primary lymph gland lobe was used. For each cell nucleus that was identified, the script automatically measured intensity in the red or green channel within a search radius around the nucleus centroid. The search radius was defined as 1.5 times the average radius of nuclei. Nuclei surrounded by above threshold intensity in the red or green channel were counted. These numbers were then used to estimate the different subsets of hemocytes in circulation.

## Ex-vivo bacterial clearance using phagocytosis assay

Hemolymph was isolated from adult flies in sterile PBS and was plated onto coverslip dishes. The adult hemocytes were allowed to attach for an hour after which the hemocytes were treated with 100 µl of diluted bacterial suspension of GFP tagged *E. coli* grown at an $OD_{600}$ = 200. The hemocytes were treated with GFP tagged *E. coli* for 30 min and the samples were then fixed with 4% paraformaldehyde. This was followed by washes with PBS and subsequent staining with Phalloidin. Images were acquired using Olympus Fluoview FV1000 confocal microscope and the images were analyzed using ImageJ to estimate the phagocytic events per hemocyte using the GFP tagged *E. coli*.

## Validation of hemocyte activation

Hemocyte activation was assessed by plating the larval circulating hemocytes derived from larvae infected with either *B. subtilis* or *E. coli*. These hemocytes were then stained with Phalloidin to study the filopodial extensions as described (*Regan et al., 2013*).

## Reporter assays to analyze humoral immune response

Toll and Imd pathway activation was assayed using *Drosomycin-GFP* and *Diptericin-GFP* reporters respectively. Expression of Drosomycin-GFP or Diptericin-GFP was assessed in the fat bodies. Toll and Imd pathway activation was tested in PSC specific Cora or NrxIV knockdown background in uninfected and infected conditions. Fat bodies were isolated, fixed and imaged using Olympus FV1000 confocal microscope.

## Statistical analysis and significance

Each experiment was performed a minimum of three times. For all the fixed tissue analysis where the corresponding lymph glands were studied using various antibodies, lymph glands from at least 10 individual wandering third instar larvae (n = 10) were analyzed. For all the live dye permeability assays, a minimum of 10 lymph gland samples (n = 10) were analyzed. For all the post infection (oral or systemic infection) assays that were done on lymph glands of larvae post infection with either *B. subtilis* or *E. coli* (including the fixed lymph gland analysis and the dye permeability assays) the number of lymph gland samples analyzed were from 10 larvae (n = 10). For the adult survival assays (post oral or systemic infection), 25 male flies (n = 25) were included in each experiment. Adult survival assays were done in triplicates and the data was pooled from the three independent experiments (N = 3). Mean average over respective total data points have been plotted in the graphs. Error bars represent standard deviation or standard error of mean as indicated in the respective figure legends. Statistical significance was determined in Microsoft Excel using unpaired *t-test* with Welch's correction. *** in the graphs represents a P-value ($p<0.001$), ** indicates $p<0.01$, * indicates $p<0.1$ and 'ns' means non-significant. For analysis of statistical significance each experimental sample was tested with respect to its respective control in a given experimental setup for all the data in each of the figures in order to estimate the P-value. Mutant genotypes were compared to the wild type controls and the knockdown or over-expression genotypes were compared to their respective parental controls for all the statistical analysis done. No statistical method was used to predetermine the sample size and the experiments were not randomized.

## Acknowledgements

We thank the Bloomington *Drosophila* Stock Center, Developmental Studies Hybridoma Bank and the fly community for fly stocks and antibodies. We thank Michael Murphy for the bacterial strains, Wayne Vogl for help with Electron Microscopy. We also thank Stephanie J Ellis and Mark Metzstein for critically reading the manuscript.

## Additional information

### Funding

| Funder | Grant reference number | Author |
| --- | --- | --- |
| Canadian Institutes of Health Research | G.T.-MOP-272122 | Guy Tanentzapf |

The funders had no role in study design, data collection and interpretation, or the decision to submit the work for publication.

### Author contributions

Rohan J Khadilkar, Conceptualization, Data curation, Formal analysis, Validation, Investigation, Methodology, Writing—original draft, Project administration, Writing—review and editing; Wayne Vogl, Formal analysis, Investigation, Methodology; Katharine Goodwin, Software, Methodology; Guy Tanentzapf, Conceptualization, Resources, Supervision, Funding acquisition, Writing—original draft, Project administration, Writing—review and editing

### Author ORCIDs

Rohan J Khadilkar, http://orcid.org/0000-0002-7297-2736
Guy Tanentzapf, http://orcid.org/0000-0002-2443-233X

### Decision letter and Author response

Decision letter https://doi.org/10.7554/eLife.28081.049
Author response https://doi.org/10.7554/eLife.28081.050

## Additional files

### Supplementary files

• Source code 1. Hemocyte counter. MATLAB source code for counting prohemocytes, differentiated cells and circulating hemocytes.
DOI: https://doi.org/10.7554/eLife.28081.046

• Source code 2. Supporting accessory MATLAB file for the hemocyte counter code file.
DOI: https://doi.org/10.7554/eLife.28081.047

• Transparent reporting form
DOI: https://doi.org/10.7554/eLife.28081.048

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
