## [Decision Letter]

Thank you for submitting your article "Modulation of occluding junctions alters the hematopoietic niche to trigger immune activation" for consideration by *eLife*. Your article has been favorably evaluated by K VijayRaghavan (Senior Editor) and three reviewers, one of whom, Yukiko M Yamashita (Reviewer #1), is a member of our Board of Reviewing Editors. The following individuals involved in review of your submission have agreed to reveal their identity: Mikio Furuse (Reviewer #2); Lucas Walzer (Reviewer #3).

The reviewers have discussed the reviews with one another and the Reviewing Editor has drafted this decision to help you prepare a revised submission.

Summary:

This manuscript reports a novel function of occluding junctions in the hematopoetic niche in the regulation of *Drosophila* prohemocyte differentiation. They bring strong evidence that the regulation of the PSC permeability barrier controls the homeostasis of the larval lymph gland.

The manuscript contains the data of great interest and proposes a novel concept not only for a role of occluding junctions in stem cell niche but also for a signaling pathway of the infection-triggered immune cell proliferation in *Drosophila*.

Overall, the experiments are carefully performed and it is clear that this manuscript describes an important discovery of high impact. Therefore, the manuscript was reviewed in favor of publication.

Essential revisions:

Despite the clear and well-controlled experiments, the text needs considerable improvement. The reviewers pointed out flaws in interpretations, description that lacks clarity and other few problems (see the bottom of the review for individual comments). Each of them is quite minor in nature (textual changes and straightforward experiments will easily address each problem), however, they should be thoroughly addressed prior to publication.

*Reviewer #1:*

This manuscript describes the discovery of permeability barrier in the PSC (the hematopoietic niche in the *Drosophila*), which is formed by septate junctions. They show that the loss of permeability barrier leads to PSC proliferation, hematopoietic progenitor proliferation and increased differentiated blood cells. Interestingly, permeability barrier is lost by infection, leading the authors to propose this is the mechanism of immune response to produce more blood cells upon infection. They further demonstrate that Toll and Imp pathways are responsible for the breakdown of permeability barrier and blood cell proliferation.

Control experiments are very carefully designed, and overall conclusion is quite exciting.

*Reviewer #2:*

This manuscript reports a novel function of occluding junctions in the hematopoetic niche in the regulation of *Drosophila* prohemocyte differentiation. The manuscript contains the data of great interest and proposes a novel concept not only for a role of occluding junctions in stem cell niche but also for a signalling pathway of the infection-triggered immune cell proliferation in *Drosophila*. The overall experiments are carefully performed. Additional data of the ultrastructure of septate junctions will improve the manuscript.

For general readers, overall organization of PSC and the localization of its septate junctions should be presented and explained in more detail.

Generally, septate junctions are located at cell-cell contacts of epithelial cells circumscribing each cell. Is PSC surrounded by a bag of epithelial cellular sheet with septate junctions? If not, do all cells in PSC have septate junctions? If the latter is correct, where are septate junctions localized in PSC cells? To clarify these issues, electron micrographs of PSC in low and middle magnification need to be presented in Figure 1. It is also important to present the ultrastructure of septate junctions in PSC by knockdown of NrxIV and Cora.

Judging from the expression of NrxIV, septate junctions in PSC seem to be pleated septate junctions, but not smooth septate junctions. To confirm this, the expressions of other pleated septate junction components ATPa, Sinu or Kune in PSC need to be confirmed.

Decreased expression of NrxIV in PSC upon bacterial infection should be presented in addition to Coracle (Figure 4).

This reviewer may not clearly understand the aim of the experiment in which SJ components were deleted in hemocytes outside of PSC, by using a pan hemocyte Gal4 driver (Results, fifth paragraph). Do hemocytes have septate junctions? Why did this experiment cause an increase in the number of cells differentiated from prohemocytes?

It is also difficult to understand "the second possible mechanism" whereby a permeability barrier in the LG can shape the PSC signalling microenvironment, based on the presented data of Wingless (Results, last paragraph).

In Figure 9', the expression and distribution of Wg should also be presented.

Based on these data, the authors need to discuss a possible mechanism for the suppression of the prohemocyte differentiation phenotype by Wg overexpression with PSC-specific Gal4.

To know whether the regulation of septate junctions upon bacterial infection is unique in PSC or not, it would be quite interesting to see whether bacterial infection leads to reduction of septate junction components NrxIV and Cora in other organs, such as foregut, hindgut, salivary gland or perineural sheath.

*Reviewer #3:*

Khadilkar et al. have made some very interesting discoveries. They bring strong evidence that the regulation of the PSC permeability barrier controls the homeostasis of the larval lymph gland. They also show that bacterial infection controls the PSC permeability barrier and, thereby, lymph gland development and the cellular immune response. There is already a plethora of results, however I have several concerns and I would like to see a much revised version of this manuscript.

First, the manuscript does not read well and should be much improved. For instance, the Abstract is awkward; the Introduction is not very well documented nor precise enough; the Results section would definitely benefit from sub-headings; there are a lot of results that are not sufficiently described; the figures (and legends) are not easy to follow and the pictures often too small; finally in the Discussion, the authors do not sufficiently put their results in the context of the pathway regulating LG/blood cell differentiation.

Second, the authors claim that the increase in plasmatocyte and crystal cell is caused by precocious prohemocyte differentiation. However, the total number of cells in the LG also increases, so the differences in plasmatocyte and crystal cell number could be due to increased proliferation of intermediates progenitors (or differentiated plasmatocytes) rather than "precocious" prohemocyte differentiation. The author must show whether the amount and proportion of prohemocyte in the primary lobe is affected.

Third and more importantly, the authors do not really explore the mechanisms by which the changes in PSC cells permeability affect primary lobe homeostasis (PSC cell number, primary lobe size, plasmatocyte and crystal cell number…). Actually the effect of the bacterial infection is surprisingly fast: in 12hours it leads to a ± 2 folds increase in PSC cell number while these cells are not proliferating in wild type third instar larvae! Also, their analysis of Dpp and Wg response is too preliminary and should be pushed forward to build a stronger model. These results should also be better discussed in light of the known function of these molecules in the LG.

Fourth, and also very importantly, the authors do not show how the disruption in the permeability barrier confer a (long term) survival advantage upon bacterial infection. Actually, the author must first test if Cora/NrxIV over-expression in the PSC increases fly susceptibility to infection (cf Figure 5: what happens to these larvae in term of survival?). Also, while they claim that the survival advantage could be due to an increase in LG-derived blood cells (as observed in the pupa), they do not show that this effect is long lasting (they should monitor the number of LG-derived hemocytes in the adult, not in the pupa), they do not test for instance if it increases bacterial clearance, and they do not demonstrate that this increase in LG-derived hemocytes is required to see a survival advantage.

---

## [Author Response]

Essential revisions:Despite the clear and well-controlled experiments, the text needs considerable improvement. The reviewers pointed out flaws in interpretations, description that lacks clarity and other few problems (see the bottom of the review for individual comments). Each of them is quite minor in nature (textual changes and straightforward experiments will easily address each problem), however, they should be thoroughly addressed prior to publication.Reviewer #2:This manuscript reports a novel function of occluding junctions in the hematopoetic niche in the regulation of Drosophila prohemocyte differentiation. The manuscript contains the data of great interest and proposes a novel concept not only for a role of occluding junctions in stem cell niche but also for a signalling pathway of the infection-triggered immune cell proliferation in Drosophila. The overall experiments are carefully performed. Additional data of the ultrastructure of septate junctions will improve the manuscript.For general readers, overall organization of PSC and the localization of its septate junctions should be presented and explained in more detail.

We now include a schematic of the PSC and where septate junctions are located. This schematic is presented in Figure 1—figure supplement 2.

Generally, septate junctions are located at cell-cell contacts of epithelial cells circumscribing each cell. Is PSC surrounded by a bag of epithelial cellular sheet with septate junctions? If not, do all cells in PSC have septate junctions? If the latter is correct, where are septate junctions localized in PSC cells? To clarify these issues, electron micrographs of PSC in low and middle magnification need to be presented in Figure 1. It is also important to present the ultrastructure of septate junctions in PSC by knockdown of NrxIV and Cora.

The reviewer asks an interesting question. The PSC is composed of cells that would not typically be considered epithelial. We found no evidence that the PSC is surrounded by a bag of epithelial cells. Rather, all the cells in the PSC seem to have septate junctions. However since PSC cells do not exhibit the conspicuous polarity of epithelial cells it seems that septate junctions are not localized to a particular domain. Instead our data suggests that in the PSC cells septate junctions are found in multiple sites, corresponding to contact sites with neighboring PSC cells, and form extensive large plaques. We completely agree that these observations should be illustrated with additional electron micrographs, specifically, low and mid magnification images requested by the reviewer and this new data are now included in Figure 1—figure supplement 2 and discussed in the subsection “Septate junctions form a permeability barrier at the PSC in the lymph gland”.

In addition, as requested by the reviewer we now include micrographs of the PSC following knockdown of Cora.These images show a striking near complete absence of SJs (Figure 1—figure supplement 2 and discussed in the subsection “Loss of Septate Junctions in the PSC results in increased prohemocyte differentiation and a higher number of cells in the PSC”).

Judging from the expression of NrxIV, septate junctions in PSC seem to be pleated septate junctions, but not smooth septate junctions. To confirm this, the expressions of other pleated septate junction components ATPa, Sinu or Kune in PSC need to be confirmed.

This is an excellent point. In order to address the reviewer’s query about the nature of septate junctions in the PSC we have analyzed the expression of the pleated septate junction components ATPalpha (using a YFP trap line), as well as Sinu and Kune-kune (both using immunofluorescence). As the reviewer suggests all three are expressed in the PSC consistent with the conclusion that the septate junctions in PSC are indeed pleated septate junctions. This new data is shown in Figure 1—figure supplement 2’ and described in the subsection “Septate junctions form a permeability barrier at the PSC in the lymph gland”.

Decreased expression of NrxIV in PSC upon bacterial infection should be presented in addition to Coracle (Figure 4).

As the reviewer requests we now include this data and it is presented in Figure 4—figure supplement 1’. Also, see the subsection “Bacterial infection disrupts the permeability barrier at the PSC triggering hemocyte differentiation”.

This reviewer may not clearly understand the aim of the experiment in which SJ components were deleted in hemocytes outside of PSC, by using a pan hemocyte Gal4 driver (Results, fifth paragraph). Do hemocytes have septate junctions? Why did this experiment cause an increase in the number of cells differentiated from prohemocytes?

As the reviewer suggests, this experiment should have been presented in a clearer way. Firstly, the evidence we show very much supports the conclusion that the expression of septate junctions proteins in the lymph gland is not restricted to the PSC (See Figure 1’). It was incorrect of us to describe the Hml-deltaGal4 as a pan-hemocyte driver. The Hml-deltaGal4 is in fact a Cortical Zone or differentiated hemocyte specific driver. As such given the context of the rest of our data using this driver is not really relevant to the story we present. We therefore chose to not include the Hml-deltaGal4 and instead present it as a part of follow up studies where we explore possible roles for SJ at later, differentiated, hemocytes.

It is also difficult to understand "the second possible mechanism" whereby a permeability barrier in the LG can shape the PSC signalling microenvironment, based on the presented data of Wingless (Results, last paragraph).

We apologize for the lack of clarity on our part, we were trying to be succinct and did not do a great job explaining this model. We have reworked our explanation of this model in the revised manuscript. See subsection “Permeability barrier breakdown at the PSC introduces localized changes in the PSC micro-environment affecting Wingless and BMP signalling”.

In Figure 9', the expression and distribution of Wg should also be presented.

This data is shown in Figure 9—figure supplement 1’’.

Based on these data, the authors need to discuss a possible mechanism for the suppression of the prohemocyte differentiation phenotype by Wg overexpression with PSC-specific Gal4.

As requested by the reviewer we have now elaborated, as much as we could given word limits on *eLife* submissions, the discussion of the possible mechanism of wg mediated suppression of differentiation elaborately in the text. See Discussion, third paragraph.

To know whether the regulation of septate junctions upon bacterial infection is unique in PSC or not, it would be quite interesting to see whether bacterial infection leads to reduction of septate junction components NrxIV and Cora in other organs, such as foregut, hindgut, salivary gland or perineural sheath.

This is an excellent suggestion and we have carried out these experiments. Specifically, we now looked at Coracle expression, before and after bacterial infection, in other organs including: the foregut, hindgut, salivary gland and fat body and found no changes in Cora level following infection. This new data is shown in Figure 4—figure supplement 2 and described in the subsection 2Bacterial infection disrupts the permeability barrier at the PSC triggering hemocyte differentiation”.

Reviewer #3:Khadilkar et al. have made some very interesting discoveries. They bring strong evidence that the regulation of the PSC permeability barrier controls the homeostasis of the larval lymph gland. They also show that bacterial infection controls the PSC permeability barrier and, thereby, lymph gland development and the cellular immune response. There is already a plethora of results, however I have several concerns and I would like to see a much revised version of this manuscript.First, the manuscript does not read well and should be much improved. For instance, the Abstract is awkward; the Introduction is not very well documented nor precise enough; the Results section would definitely benefit from sub-headings; there are a lot of results that are not sufficiently described; the figures (and legends) are not easy to follow and the pictures often too small; finally in the Discussion, the authors do not sufficiently put their results in the context of the pathway regulating LG/blood cell differentiation.

We thank the reviewer for the very helpful suggestions and advice. We have made a substantial effort to improve upon the weaknesses they identified in the text. Some examples: we have rewritten the part of the Abstract the reviewer found problematic, we rewrote and expanded parts of the Introduction to make it more precise and better documented (See second, third and last paragraphs). We have tried to expand the descriptions in the Results to the best of our abilities given the word limit in *eLife* (see subsection “Loss of Septate Junctions in the PSC results in increased prohemocyte differentiation and a higher number of cells in the PSC”; subsection “Permeability barrier breakdown at the PSC introduces localized changes in the PSC micro-environment affecting Wingless and BMP signalling”).

We have substantially reworked our figures to make the images bigger, the information more compact, and the legends easier to follow. (e.g. Figure 4, Figure 2—figure supplement 1–Figure 2—figure supplement 5, Figure 4—figure supplement 1–Figure 4—figure supplement 3, Figure 5—figure supplement 1 and Figure 5—figure supplement 2).

Finally, as requested, we rewrote and expanded the Discussion to put our results in context of the pathways regulating LG/blood cell differentiation (see Discussion, second and third paragraphs).

Second, the authors claim that the increase in plasmatocyte and crystal cell is caused by precocious prohemocyte differentiation. However, the total number of cells in the LG also increases, so the differences in plasmatocyte and crystal cell number could be due to increased proliferation of intermediates progenitors (or differentiated plasmatocytes) rather than "precocious" prohemocyte differentiation. The author must show whether the amount and proportion of prohemocyte in the primary lobe is affected.

The reviewer raised an important point that we have addressed with additional experiments. Specifically, we sought to address whether the differences in plasmatocyte and crystal cell number is due to an increase in prohemocyte differentiation or due to increased proliferation of intermediates progenitors. To accomplish this we used the reporter domeMESOlacz as a marker for prohemocyte fate (Hombria et al., 2005; Gao et al., 2009; Tokusumi et al., 2012; Oyallon et al., 2016). This combined with DAPI and plasmatocyte marker, P1 allowed us to quantify the total number of prohemocytes, the total number of plasmatocytes (representative of differentiated hemocytes) and the total number of cells in the lymph gland, as well the ratio of prohemocyte numbers to the total numbers of cells in the LG. Overall these results strongly support our claim that the increase in plasmatocyte and crystal cells that we see following loss of the permeability barrier is caused by prohemocyte differentiation. We found that following SJ depletion in the PSC the total number of prohemocytes is decreased while the total number of plasmatocytes, the ratio of plasmatocytes to total cells in the LG and the total number of cells in the lymph gland is increased. This additional data is shown in Figure 2—figure supplement 5 and discussed in the subsection “Loss of Septate Junctions in the PSC results in increased prohemocyte differentiation and a higher number of cells in the PSC”.

Third and more importantly, the authors do not really explore the mechanisms by which the changes in PSC cells permeability affect primary lobe homeostasis (PSC cell number, primary lobe size, plasmatocyte and crystal cell number…). Actually the effect of the bacterial infection is surprisingly fast: in 12hours it leads to a ± 2 folds increase in PSC cell number while these cells are not proliferating in wild type third instar larvae! Also, their analysis of Dpp and Wg response is too preliminary and should be pushed forward to build a stronger model. These results should also be better discussed in light of the known function of these molecules in the LG.

We agree with the reviewer that we needed to explore in greater depth how changes in the permeability barrier cause such profound and dynamic response in LG homeostasis. We now include a couple of additional experiments to provide more in-depth analysis. As the reviewer asserts, there is a rapid and dramatic increase in the number of PSC cells, which is puzzling given that these cells are not proliferating. To our eyes at least the situation we encounter here is similar to that we observed in our recent exploration of the effects of permeability barrier removal upon another stem cell niche model found in the *Drosophila* testes (see Fairchild et al., Current Biology, 2016). In the testes removal of a permeability barrier also resulted in rapid niche expansion even though niche cells are generally not thought of as being mitotic. Using lineage-tracing approaches we showed that this was due to reprogramming and subsequent recruitment into the niche of differentiated cells, likely as a result of changes in the signaling environment following loss of the permeability barrier. We are very curious to carry out a similar lineage tracing experiment in the LG to see if something similar is going on. However as these experiments are very time consuming we believe they are outside of the scope of the current work. Nonetheless, we added a number of pieces of data that help clarify how the signaling environment in the PSC is altered following depletion of the SJ:

1) One mechanism that is known to underlie the sort of increase in PSC cell number that we observe following depletion of SJs, is altered BMP signaling. Specifically, it is known that the heparan sulfate proteo-glycan, Dally like protein (Dlp), is highly expressed in the PSC where it regulates the levels of BMP signalling (Pennetier et al., 2012). Loss of Dlp causes increased BMP signaling and expansion in the number of PSC cells (Pennetier et al., 2012). We now report data showing the depletion of SJ components results in lower Dlp levels in the PSC which helps explain the expansion phenotype. This new data is shown in Figure 9—figure supplement 2’, G and described in the subsection “Permeability barrier breakdown at the PSC introduces localized changes in the PSC micro-environment affecting Wingless and BMP signalling”.

2) We explored another known mechanism that regulates PSC cell size involving dMyc. It was previously shown that dMyc is normally repressed in the PSC by BMP signaling, loss of BMP from the PSC results in dMyc de-repression and consequently PSC size increase (Pennetier et al., 2012). We have added new data to the manuscript showing dMyc is upregulated in the PSC following depletion of SJs. Moreover, knockdown of dMyc in the PSC concomitant with SJ depletion abrogates PSC expansion. This analysis implicates dMyc and its downstream targets in the pathway that leads to PSC expansion following SJ depletion. This new data is shown in Figure 9—figure supplement 2’, H and Figure 9—figure supplement 2 and described in the subsection “Permeability barrier breakdown at the PSC introduces localized changes in the PSC micro-environment affecting Wingless and BMP signalling”.

3) In addition to this, we also tested if Coracle expression at the PSC is itself downstream of Wg or BMP inactivation. We found that Cora is expressed independently of either pathway suggesting a rather straightforward pathway between depletion of SJ components and alterations in the Wg and BMP pathway. This new data is shown in Figure 9—figure supplement 3 and described in the subsection “Permeability barrier breakdown at the PSC introduces localized changes in the PSC micro-environment affecting Wingless and BMP signalling”.

4) We modified the Discussion to talk about these results in a more direct and full way. These changes are found in the third paragraph of the Discussion.

Fourth, and also very importantly, the authors do not show how the disruption in the permeability barrier confer a (long term) survival advantage upon bacterial infection. Actually, the author must first test if Cora/NrxIV over-expression in the PSC increases fly susceptibility to infection (cf Figure 5: what happens to these larvae in term of survival?). Also, while they claim that the survival advantage could be due to an increase in LG-derived blood cells (as observed in the pupa), they do not show that this effect is long lasting (they should monitor the number of LG-derived hemocytes in the adult, not in the pupa), they do not test for instance if it increases bacterial clearance, and they do not demonstrate that this increase in LG-derived hemocytes is required to see a survival advantage.

The reviewer raises two excellent points and we have now carried out additional experiments to address both comments in full:

1) We have now tested whether Cora/NrxIV over-expression in the PSC, which we might expect to reduce the flies’ ability to induce the cellular immune response, increases fly susceptibility to infection. However, we did not find increased susceptibility, suggesting that even if we compromise the cellular immune response, flies can defend themselves from pathogens. This may be accomplished by compensatory action of the humoral immune response using Anti Microbial Peptides (AMPs). This new data is shown in Figure 5—figure supplement 2 and described in the subsection “Flies bearing septate junction depletion in the PSC mount a robust hemocyte-mediated cellular immune response leading to better survival upon infection”.

2) In order to properly show “long-lasting” survival advantage following depletion of SJ component from the larval PSC we performed representative hemocyte counts in control and SJ depleted adult flies. We consistently found higher hemocyte numbers in adults following SJ depletion in the larva. This new data is shown in Figure 6’’ and described in the subsection “Flies bearing septate junction depletion in the PSC mount a robust hemocyte-mediated cellular immune response leading to better survival upon infection”.

Higher hemocyte numbers might not, by themselves, confer a survival advantage. Therefore we assayed directly the function of the cellular immune response following SJ depletion from the larval PSC. Specifically, we assayed in vivo bacterial clearance and an ex vivo bacterial phagocytosis. Both showed improved immune system function following larval depletion of SJs. This new data is shown in Figure 4—figure supplement 3 and described in the subsection “Flies bearing septate junction depletion in the PSC mount a robust hemocyte-mediated cellular immune response leading to better survival upon infection”. Taken together these data clearly demonstrate how effects on cellular immunity can extend into adult life and, in that sense, provide a “long-lasting” advantage.